# Purified F-ATP synthase forms a Ca$^{2+}$-dependent high-conductance channel matching the mitochondrial permeability transition pore

Andrea Urbani [1], Valentina Giorgio [1,2], Andrea Carrer[1], Cinzia Franchin[1,3], Giorgio Arrigoni[1,3], Chimari Jiko[4], Kazuhiro Abe [5], Shintaro Maeda[6], Kyoko Shinzawa-Itoh[7], Janna F.M. Bogers[8], Duncan G.G. McMillan [8], Christoph Gerle [9,10]*, Ildikò Szabò[2,11]* & Paolo Bernardi [1,2]*

The molecular identity of the mitochondrial megachannel (MMC)/permeability transition pore (PTP), a key effector of cell death, remains controversial. By combining highly purified, fully active bovine F-ATP synthase with preformed liposomes we show that Ca$^{2+}$ dissipates the H$^{+}$ gradient generated by ATP hydrolysis. After incorporation of the same preparation into planar lipid bilayers Ca$^{2+}$ elicits currents matching those of the MMC/PTP. Currents were fully reversible, were stabilized by benzodiazepine 423, a ligand of the OSCP subunit of F-ATP synthase that activates the MMC/PTP, and were inhibited by Mg$^{2+}$ and adenine nucleotides, which also inhibit the PTP. Channel activity was insensitive to inhibitors of the adenine nucleotide translocase (ANT) and of the voltage-dependent anion channel (VDAC). Native gel-purified oligomers and dimers, but not monomers, gave rise to channel activity. These findings resolve the long-standing mystery of the MMC/PTP and demonstrate that Ca$^{2+}$ can transform the energy-conserving F-ATP synthase into an energy-dissipating device.

[1] Department of Biomedical Sciences, University of Padova, 35131 Padova, Italy. [2] Consiglio Nazionale delle Ricerche Neuroscience Institute, 35131 Padova, Italy. [3] Proteomics Center, University of Padova and Azienda Ospedaliera di Padova, 35131 Padova, Italy. [4] Institute for Integrated Radiation and Nuclear Science, Kyoto University, Kyoto, Japan. [5] Cellular and Structural Physiology Institute, Nagoya University, Nagoya, Japan. [6] Department of Integrative Structural and Computational Biology, Scripps Research Institute, La Jolla, CA 92037, USA. [7] Department of Life Science, Graduate School of Life Science, University of Hyogo, Hyogo 678-1297, Japan. [8] Department of Biotechnology, Delft University of Technology, 2629 HZ Delft, The Netherlands. [9] Institute for Protein Research, Osaka University, Suita, Osaka, Japan. [10] Japan Agency for Medical Research and Development (AMED), Tokyo, Japan. [11] Department of Biology, University of Padova, Padova, Italy. *email: gerle.christoph@protein.osaka-u.ac.jp; ildi@bio.unipd.it; bernardi@bio.unipd.it

The permeability transition (PT) is a $Ca^{2+}$-dependent permeability increase of the mitochondrial inner membrane leading to depolarization and cessation of ATP synthesis. The cutoff of about 1500 Da allows inner membrane permeabilization to ions and solutes with matrix swelling. Known since the early days of research on isolated organelles, the molecular basis of the PT has remained a mystery for >60 years[1]. The PT is mediated by opening of a channel, the permeability transition pore (PTP), also called mitochondrial megachannel[2] (MMC) or multiconductance channel[3] because of its electrophysiological properties[4]. Following the observation that the PTP activator cyclophilin (CyP) D interacts with the F-ATP synthase in a cyclosporin (Cs)A-sensitive manner[5] (which matches MMC/PTP activation by CyPD and desensitization by CsA), the possibility that F-ATP synthase generates channels has been investigated with encouraging results[6–12]. Given that no obvious structural features would predict that the F-ATP synthase can form channels[13–20] and that a tightly coupled F-ATP synthase is essential to power ATP generation[21], this hypothesis may appear unlikely. However, recent cryo-EM studies have revealed a far more complex structural organization than anticipated both for the yeast dimeric $F_O$ domain[18] and for the porcine F-ATP synthase tetramer[20]. Furthermore, F-ATP synthases from a variety of sources catalyze hydrolysis of $Ca^{2+}$-ATP[22–25], yet no proton gradient forms during ATP hydrolysis[22,24,25], a finding that is compatible with $Ca^{2+}$-dependent opening of a dissipative pathway. In previous studies a solid link was established between F-ATP synthase and formation of the MMC/PTP through detection of currents upon reconstitution into planar bilayers of F-ATP synthase extracted from mitochondria or from native gels[7–10]. In spite of extensive pharmacological evidence pointing at the F-ATP synthase as being responsible for current formation, the conclusions of these studies are limited by the presence of additional proteins. To overcome these limitations we have used two complementary strategies. In the first, we have produced selective mutants of F-ATP synthase and assessed the consequences of the mutations on $Ca^{2+}$-sensitivity, inhibition by $H^+$ and modulation by specific reagents of the MMC/PTP[11,12,26–28]. In the second strategy, the results of which are described in the present manuscript, we have employed highly pure and stable F-ATP synthase from large-scale preparations from bovine hearts (Supplementary Fig. 1). These preparations, which displayed oligomycin-sensitive ATPase activity and catalyzed $H^+$ translocation after incorporation into preformed liposomes, were used to address the question of whether F-ATP synthase can give rise to bona fide channels after incorporation into planar lipid bilayers, and whether these channels match the known features of the MMC/PTP.

## Results

**Characterization of bovine F-ATP synthase.** F-ATP synthase was purified from bovine heart mitochondria by a combination of sucrose density gradient centrifugation and ion-exchange chromatography employing the mild, high-affinity detergent lauryl-maltose-neopentyl glycol (LMNG)[15,29,30]. Clear-native gel electrophoresis documented the presence of monomeric, dimeric and oligomeric forms of the enzyme complex (Fig. 1a), which were also visualized by negative stain EM (Fig. 1b). Denaturing sodium dodecyl sulfate (SDS) gel electrophoresis (Fig. 1c) and mass spectrometry (Supplementary Tables 1 and 2) allowed detection of all the expected subunits, including the labile 6.8PL and DAPIT. The preparation contained very low amounts of contaminating proteins (Supplementary Table 2; see Data Availability for access to mass spectrometry proteomics data) and catalyzed oligomycin-sensitive ATP hydrolysis (Fig. 1d, e) with the typical

concentration-dependence on ATP and $Mg^{2+}$ (Supplementary Fig. 3a, b). In these measurements we used the standard assay based on coupled enzymatic reactions, which cannot be utilized to study $Ca^{2+}$-ATPase activity because of the $Mg^{2+}$ requirement of pyruvate kinase (Supplementary Fig. 3c, d). With a method based on direct detection of Pi release it became clear that the bovine F-ATP synthase was also capable to hydrolyze $Ca^{2+}$-ATP (Supplementary Fig. 4). The total amount of phospholipids per F-ATP synthase monomer in the Poros column fractions relevant to this study were $150 \pm 54$ ($n = 8$). Of these, $57 \pm 5\%$ was phosphatidylethanolamine, $16 \pm 2\%$ was phosphatidylcholine and $27 \pm 1\%$ was cardiolipin. The latter value compares quite well with the in situ cardiolipin content of the inner mitochondrial membrane of about 18%[31].

This preparation was then incubated with preformed liposomes, a procedure that yielded high density of incorporation, as detected by both cryo-EM (Fig. 1f) and negative stain EM (Fig. 1, compare g and h). Morphometric analysis of (proteo)liposomes revealed a clear preference for incorporation into smaller liposomes, with a distribution centered at a diameter of about 40 nm (Fig. 1i, j). Re-solubilization of the F-ATP synthase proteoliposomes with digitonin and subsequent analysis by BN-PAGE indicates the presence of monomeric, dimeric and oligomeric F-ATP synthase (Supplementary Fig. 5). After reconstitution into proteoliposomes F-ATP synthase was catalytically active and sensitive to oligomycin (Fig. 2a); and upon addition of 2.5 mM ATP it catalyzed $H^+$ translocation with the establishment of a $H^+$ gradient, as assessed by 9-amino-6-chloro-2-methoxyacridine (ACMA) quenching (Fig. 2b, c). The $H^+$ gradient was stable and could be rapidly dissipated by the addition of nigericin (Fig. 2c). Given that $Mg^{2+}$ and adenine nucleotides are effective inhibitors of the PTP, we also performed these experiments with 0.125 mM $Mg^{2+}$ATP for testing the effect of PTP inducers after proton pumping. Addition of $Ca^{2+}$ alone slightly perturbed the $H^+$ gradient (Fig. 2d), while the PTP agonists benzodiazepine (Bz)-423 and phenylarsine oxide (PhAsO) caused a slow, partial release of the $H^+$ gradient (Fig. 2e), which is probably due to inhibition of F-ATP synthase by Bz-423[32]. Bz-423 is a small organic compound (MW 441 Da) that was discovered in a screening of molecules able to selectively kill autoreactive B lymphocytes[33] and later shown to selectively bind F-ATP synthase in a phage display assay[32]. Since Bz-423 binds F-ATP synthase subunit OSCP[34] at the same site as CyPD, resulting in MMC/PTP activation[7], in this study we have used Bz-423 rather than CyPD as the standard agonist because of its availability, stability and lack of interference with the experimental set-up. PhAsO is a dithiol cross-linker that mimicks the effect of oxidants[35] and has been extensively characterized in studies of the PT in mitochondria[1]. When 2 mM $Ca^{2+}$ was also added to the proteoliposomes, together with Bz-423 and PhAsO, immediate and complete collapse of the $H^+$ gradient was observed (Fig. 2f). Thus, when treated with well-characterized MMC/PTP inducers in the presence of $Ca^{2+}$, F-ATP synthase proteoliposomes energized via ATP hydrolysis-powered acidification undergo full permeabilization, suggestive of MMC/PTP opening.

**Channel formation by F-ATP synthase.** Next, we tested whether F-ATP synthase forms channels after fusion of the proteoliposomes into planar lipid bilayer membranes. Since $Ca^{2+}$ alone is sufficient to elicit MMC activity in patch clamp experiments in mitoplasts[4], we firstly addressed the key question of whether $Ca^{2+}$ is necessary and sufficient to trigger channel activity by F-ATP synthase. Proteoliposomes were added to the cis chamber, followed by a period of up to 5 min of recording, during which no

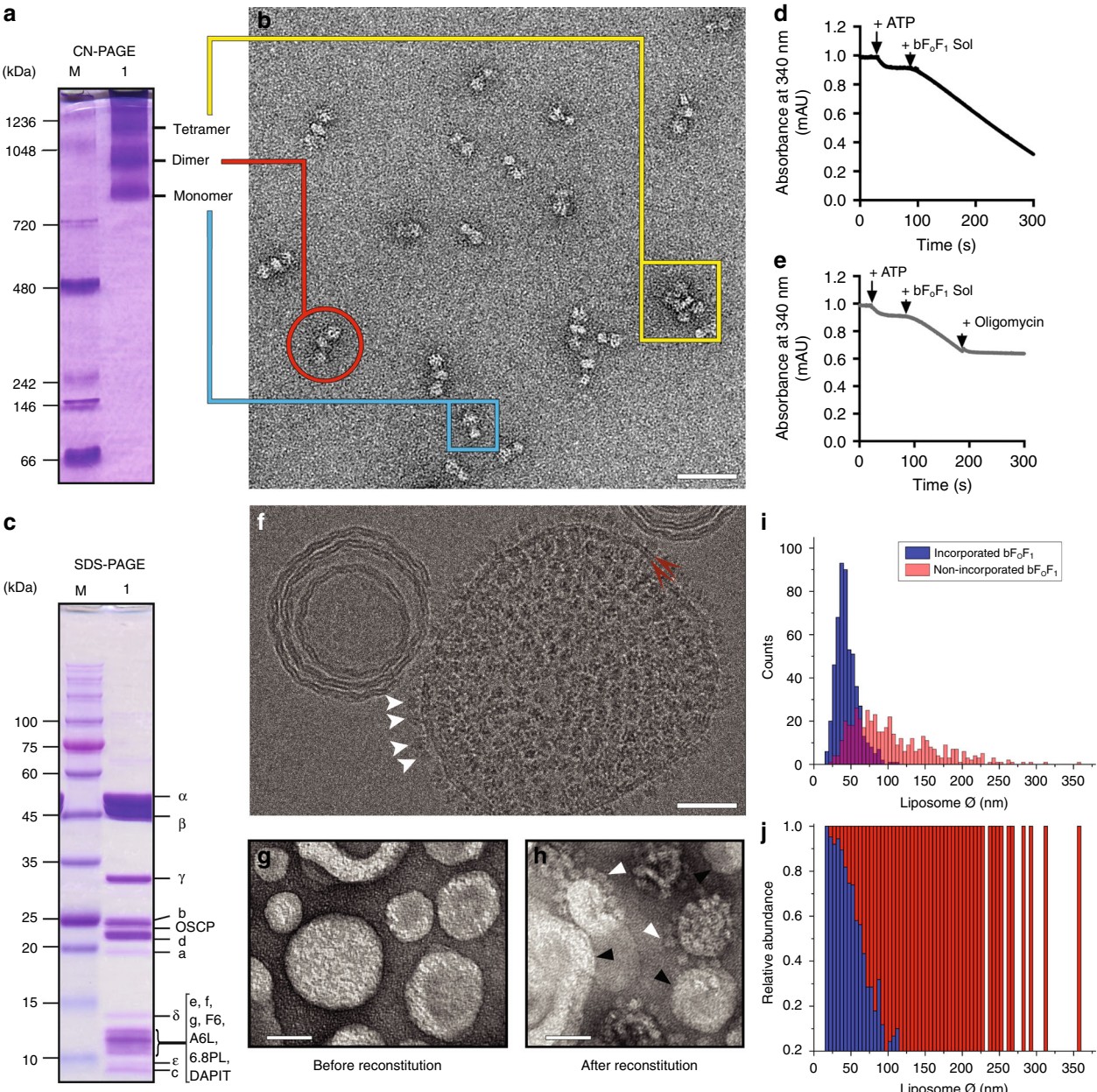

**Fig. 1** Characterization of bovine F-ATP synthase. **a** Clear-native gel electrophoresis indicates the presence of monomeric, dimeric, and oligomeric bovine F-ATP synthase, and the absence of subcomplexes or other smaller protein complexes. Lane M: molecular weight markers (kDa); Lane 1: F-ATP synthase (30 μg). **b** Negative stain EM documents the presence of bovine F-ATP synthase monomers (blue rectangle), dimers (red circle), and tetramers (yellow rectangle). Scale bar 50 nm. **c** Subunit composition was determined by denaturing SDS gel electrophoresis. All expected subunits were detected, including the very weakly associated 6.8PL and DAPIT, see also data from mass spectrometry (Supplementary Table 1). Lane M: molecular weight markers (kDa); Lane 1: F-ATP synthase (30 μg). **d**, **e** NADH oxidation-coupled enzymatic assay showing that soluble F-ATP synthase (bF$_O$F$_1$ Sol, 10 μg added where indicated) has ATPase activity (addition of ATP 2.5 mM where indicated) that can be fully inhibited by oligomycin A (2 μM where indicated). Traces are representative of three independent experiments. **f** Cryo-EM image of a bovine F-ATP synthase proteoliposome. The clearly visible tram-track features (red double arrow) confirm the formation of lipid bilayers, and F$_1$ domains protruding out of the membrane are easy to identify at the edge of the proteoliposome (white arrowheads). Scale bar 50 nm. **g** Negative stain EM of liposomes before and **h** after reconstitution of bovine F-ATP synthase. After reconstitution, F$_1$ domains (open triangles) can be clearly distinguished from naked liposomes (closed triangles). Scale bar, 50 nm. **i**, **j** Nine-hundred and ninety-nine liposomes from 19 electron microscopy images taken after incorporation of F-ATP synthase were scored for the presence of F$_1$ and their size measured with the aid of ImageJ software. Source data for panels **d**, **e**, **i**, **j** are provided as a Source Data file

currents were seen. In the experiment reported here, 0.1 mM Ca$^{2+}$ was then added to the *cis* side, and a small-conductance channel activity was detected, the amplitude of which increased as the Ca$^{2+}$ concentration was raised to 1 mM (Fig. 3a). Addition of 1 mM EGTA abolished channel openings, which were restored by the subsequent addition of Ca$^{2+}$ in excess of EGTA, similarly to

MMC measured in the native IMM[36]. It should be noted that we did not apply an osmotic gradient, which promotes fusion of proteoliposomes to the planar lipid bilayer[37]; and that liposomes with diameter lower than 100 nm have a low probability of fusion[38]. Thus, in spite of the large amount of F-ATP synthase incorporated into the liposomes, in the electrophysiological

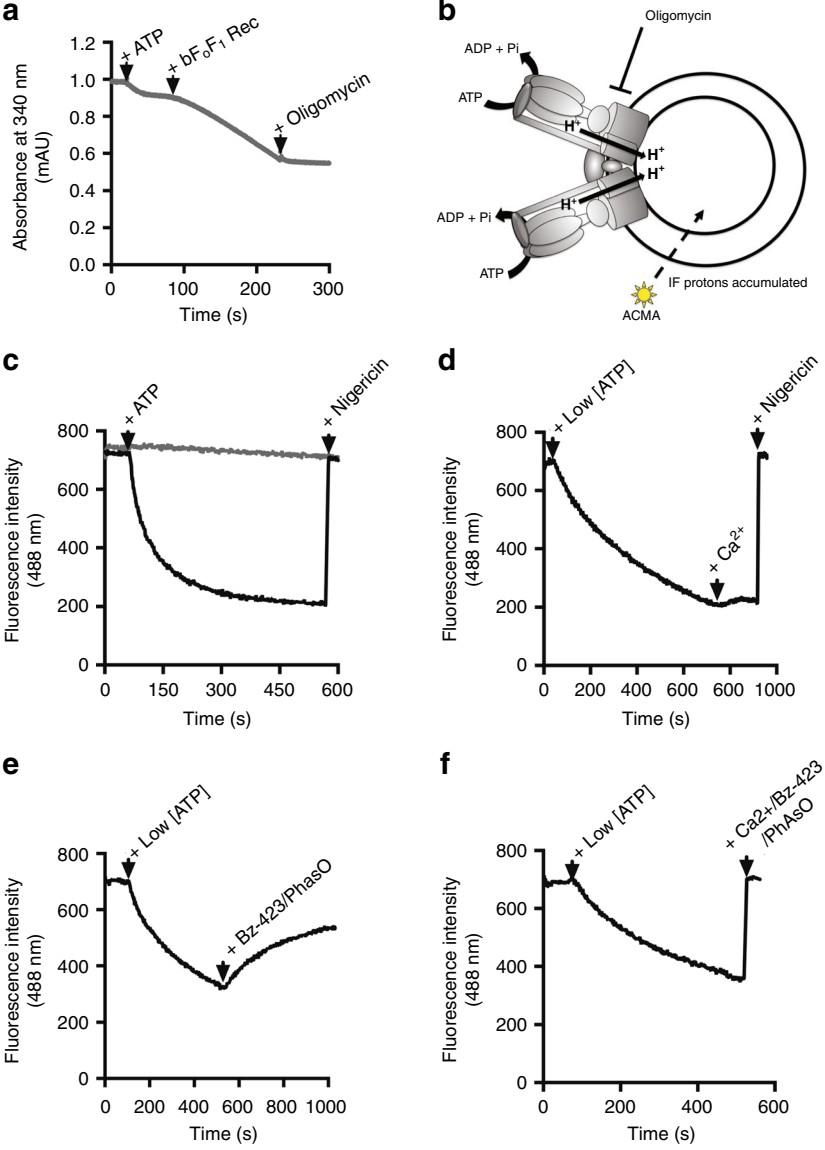

**Fig. 2** Release of the H$^+$ gradient of F-ATP synthase proteoliposomes by Ca$^{2+}$ and PTP activators. **a** ATP hydrolysis by liposome-reconstituted F-ATP synthase was measured with an ATP-regenerating assay, as described in Methods by monitoring absorbance at 340 nm. ATP (2.5 mM), 10 μg proteoliposomes (bF$_O$F$_1$ Rec), and 2 μM oligomycin were added where indicated. **b** Scheme of the ACMA-quenching assay involving F$_O$F$_1$ complexes within the proteoliposome membrane. Protons pumped into the vesicle lumen by ATP hydrolysis cause the accumulation (and quenching) of ACMA fluorescence. Oligomycin prevents ATP hydrolysis as shown in panel **a**. **c** Where indicated 2.5 mM ATP and 1 μM Nigericin; gray trace, 2 μM oligomycin was present. **d** Where indicated 0.125 mM Mg$^{2+}$ATP, 2 mM Ca$^{2+}$, and 1 μM Nigericin. **e** Where indicated 0.125 mM Mg$^{2+}$ATP, 150 μM Bz-423, and 60 μM PhAsO. **f** Where indicated 0.125 mM Mg$^{2+}$ATP, 2 mM Ca$^{2+}$, 150 μM Bz-423, and 60 μM PhAsO. Each trace is representative of three similar experimental replicates. Source data for panels **a**, **c**–**f** are provided as a Source Data file

recordings we observed few events, which is a desirable feature if single-channel activity is to be assessed. To further explore the frequency of fusion events, we also used proteoliposomes of the same size, composition, and lipid-to-protein ratio where grami-cidin D (which forms channels with high efficiency) had been incorporated with the same protocol used for the F-ATP syn-thase. Although the two proteins obviously differ substantially, also in this case activity of well-resolved single channels could be detected (Supplementary Fig. 6). Overall, with F-ATP synthase proteoliposomes we observed channel activity in 13 out of 16 experiments (81%) and with gramicidin D in 6 out of 8 (75%).

Given that in these protocols the majority of F-ATP synthase molecules appears to remain in proteoliposomes not fused to the planar membrane, in subsequent electrophysiological

experiments we have mostly added the purified, LMNG-stabilized preparation directly to the *cis* chamber, a procedure that yielded channel activity in 103 out of 131 experiments (79%). In addition to Ca$^{2+}$, a variety of additional inducers, such as Bz-423 and PhAsO are required for PTP opening and for detection of high-conductance channels in gel-purified preparations[7,9,10]. After confirming ohmic resistance of the lipid bilayer, we added 3 mM Ca$^{2+}$, 0.1 mM PhAsO, and 0.2 mM Bz-423, conditions that elicited complex channel activity with a maximum conductance of about 1.3 nS and multiple conductance substates (Fig. 3b and Supplementary Fig. 7). As already mentioned, these substates are typical of the channel in the native membrane[2,3], and are consistent with the alternative denomination of the MMC as mitochondrial multiconductance channel[3]. Activity could also be

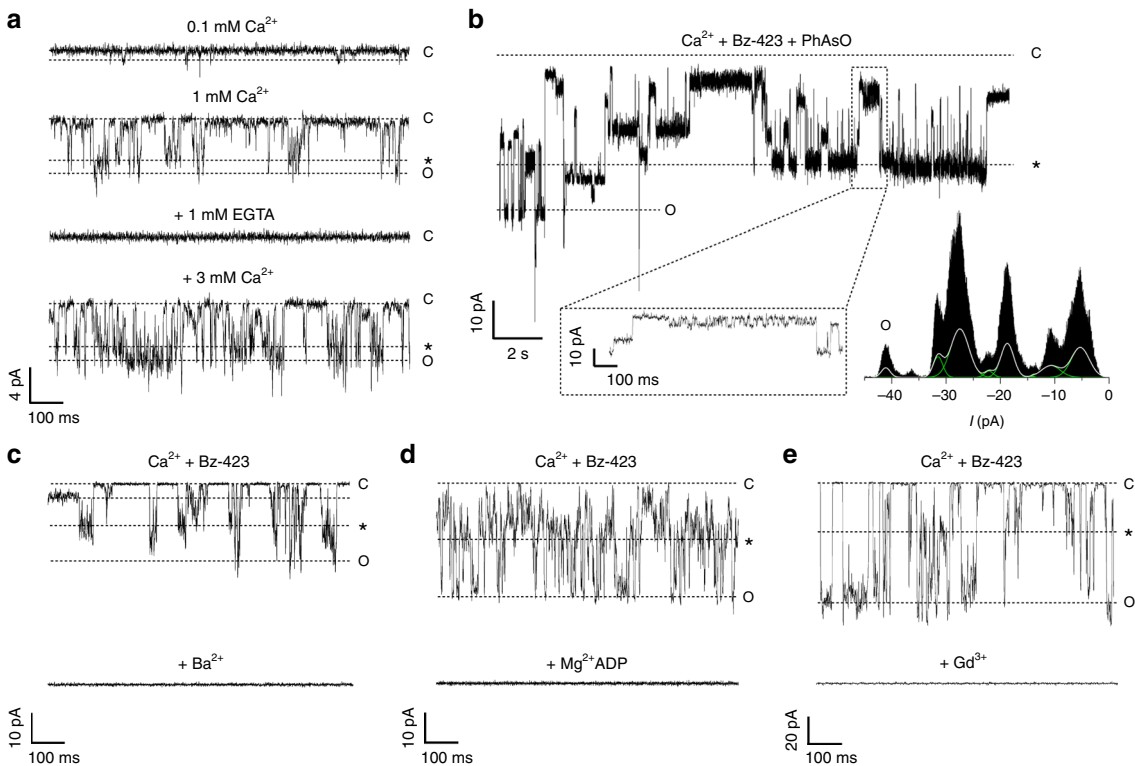

**Fig. 3** Activation and inhibition of bovine F-ATP synthase channels. **a** F-ATP synthase channel activity was elicited after reconstitution of the protein into proteoliposomes and insertion in the bilayer in the presence of increasing concentrations of $Ca^{2+}$. Addition of EGTA suppressed channel activity, which was reactivated by the further addition of $Ca^{2+}$. Conductance at 0.1 mM $Ca^{2+}$ was $30 \pm 10$ pS and at 1–3 mM $Ca^{2+}$ was $117 \pm 17$ pS (SEM, $n = 3$); $V_{cis} = -60$ mV. **b** Representative current trace after direct incorporation of F-ATP synthase showing long lasting and complex channel activity after addition of 3 mM $Ca^{2+}$, 0.1 mM Bz-423, and 0.2 mM PhAsO; $V_{cis} = -80$ mV. The closed state is denoted with C, the most frequent subconductance state with an asterisk, and the maximal open state with O. Inset, low conductance (30 pS) activity can be observed in the same trace, together with the higher conductance states. Multiple current levels are distinguishable as different peaks in the current amplitude histogram shown below the current tracings (see also Supplementary Fig. 7a from the same experiment). **c–e** Current traces were recorded after direct incorporation of F-ATP synthase and addition of increasing concentrations of $Ca^{2+}$ up to 3 mM in the presence of 0.15 mM Bz-423; after a stable channel activity was reached, inhibitors were added. From the left: current traces before and after addition of **c** 5 mM $Ba^{2+}$ ($[Ca^{2+}] = 3$ mM, $G_{max} = 500$ pS); **d** 1 mM $Mg^{2+}$ADP ($[Ca^{2+}] = 0.3$ mM, $G_{max} = 600$ pS); and **e** 1 mM $Gd^{3+}$ ($[Ca^{2+}] = 0.66$ mM, $G_{max} = 1300$ pS). Traces are representative of at least nine replicates for each inhibitor. The source data underlying panel **a** are provided as a Source Data file

elicited within about 3 min of the addition of $Ca^{2+}$ in the presence of Bz-423 only (Supplementary Fig. 8). Channel activity was inhibited by the well-known MMC/PTP inhibitors $Ba^{2+}$ (Fig. 3c and Supplementary Fig. 9a) and $Mg^{2+}$ADP (Fig. 3d and Supplementary Fig. 9b) as well as by $GdCl_3$ (Fig. 3e and Supplementary Fig. 9c), a general cation channel blocker able to inhibit MMC/PTP in mitoplasts[12].

We next analyzed the current–voltage relationship of the reconstituted F-ATP synthase channels. Bz-423 and $Ca^{2+}$ were added; after a stable activity had been reached, currents were recorded at $V_{cis}$ ranging from $-80$ to $+20$ mV. An essentially linear current–voltage relationship was observed (Fig. 4a) with an increased propensity of the channels for open states at more depolarized voltages (Supplementary Fig. 10). To test whether F-ATP synthase insertion in the planar membrane occurred with a preferential orientation, we added it to the *cis* side while $Ca^{2+}$ (up to 3 mM) and 0.15 mM Bz-423 were added to the *trans* side. No channel activity was detected, while the subsequent addition of the same agonists to the *cis* side readily elicited channel activity (Fig. 4b). This set of findings indicates that the F-ATP synthase inserts into the membrane with its hydrophobic $F_O$ domain, leaving the large hydrophilic $F_1$ sector accessible from the *cis* side as observed in the proteoliposomes (Fig. 1f, h). We also examined the electrophysiological behavior of samples cut out from BN-

PAGE. As was the case for proteins eluted from gels after separation of total mitochondrial extracts[7], currents were detectable after insertion of proteins extracted from dimer and oligomer, but not monomer bands (Fig. 4c). Maximal conductance of channels formed by dimers and oligomers was not significantly different (Supplementary Fig. 11). Channel activity was observed in 9 out of 10 experiments for oligomers (90%), 8 out of 9 for dimers (89%), and 0 out of 7 (0%) for monomers.

The ANT and VDAC have long been suspected to participate in MMC/PTP formation, also based on their channel-forming activity assessed by electrophysiology[39–41]. Given that our preparation contains low amounts of both proteins (Supplementary Table 2) we tested the effect of bongkrekate (BKA) (Supplementary Fig. 12a) and of König's polyanion (Supplementary Fig. 12b), the selective inhibitors of ANT[42] and of VDAC[40], respectively. No inhibition was observed in either case, demonstrating that neither ANT nor VDAC contributes to MMC/PTP-like activity of membrane-reconstituted F-ATP synthase.

We also tested if Bz-423 is capable of stabilizing the $Ca^{2+}$-activated channel in its full-conductance state. Reliable single-channel current analysis is negatively affected by flickering and by the presence of multiple current levels, typical MMC behaviors

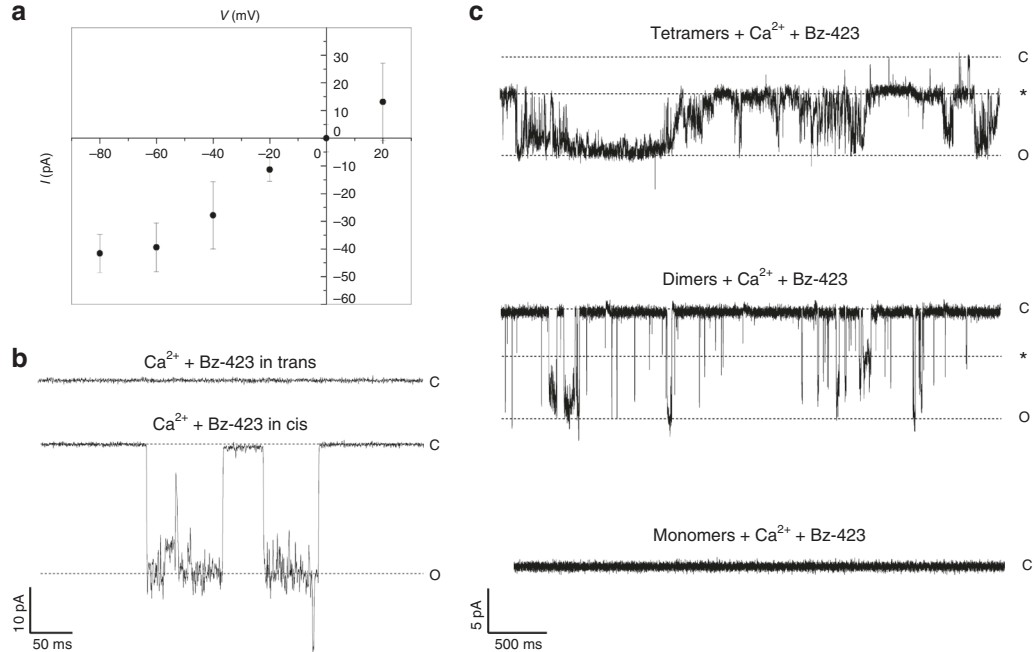

**Fig. 4** Characterization of channel activity of bovine F-ATP synthase. **a** Currents were recorded at the indicated voltages ($n = 7$, error bars refer to SEM). **b** F-ATP synthase was added to the *cis* side of the recording cuvette. No channel activity was detected when $Ca^{2+}$ up to 3 mM and 0.15 mM Bz-423 were added to the *trans* side, while channel activity appeared when the same concentrations of $Ca^{2+}$ and Bz-423 were subsequently added to the *cis* side ($V_{cis} = -60$ mV, $G = 500$ pS for prevalent conductance state indicated with O). The two traces shown are from one representative experiment of four. **c** Electrophysiological assessment of gel-eluted bovine F-ATP synthase. Monomers, dimers, and tetramers of F-ATP synthase were eluted from BN-PAGE gels and reconstituted into planar lipid bilayers. Channel activity was assessed in the presence of $Ca^{2+}$ up to 3 mM and 0.15 mM Bz-423 added to the *cis* side. Both dimers ($n = 8$) and tetramers ($n = 9$) elicited channel activity while monomers elicited no activity ($n = 7$). Representative traces are shown at $V_{cis} = -60$ mV. The closed state is denoted with C, the most frequent subconductance state with an asterisk, and the maximal open state with O. The source data underlying panel **a** are provided as a Source Data file

that are both observed in our recordings. To overcome this potential problem, we measured the power spectrum area (PSA), which provides a generally applicable estimate of channel activity based on the integral of the power spectrum over the entire frequency range of the acquired current signal. We developed an algorithm (Supplementary Tables 3 and 4) that allows to evaluate total channel activity over a long (10-min) recording time, and calculated the corresponding PSA, open probability (Po), and conductance. These experiments revealed that Bz-423 was not able to trigger channel opening per se, yet it allowed channels to reach a conductance matching that of PTP/MMC when $Ca^{2+}$ was also present (Fig. 5a–d and Supplementary Fig. 13). A synergistic effect is also supported by the cooperativity observed for both Po and PSA when the $Ca^{2+}$ concentration was raised in the presence of a fixed concentration of Bz-423, consistent with the Hill coefficient between 1 and 2 obtained from data analysis (Fig. 5e). Mean and maximal conductance increased as a function of the $Ca^{2+}$ concentration (Fig. 5f), suggesting that $Ca^{2+}$ may contribute to channel assembly and/or trigger specific conformational changes.

## Discussion
In this manuscript, we have provided solid evidence that, after treatment with $Ca^{2+}$, highly purified bovine F-ATP synthase can give rise to high-conductance channels. Channel openings are stabilized by MMC/PTP agonists and inhibited by MMC/PTP inhibitors. These results are consistent with previous electrophysiological work[7–10], but represent a key step forward because of the purity, completeness and intactness of the bovine F-ATP synthase complexes. Of specific importance is the demonstration

that the preparation displays oligomycin-sensitive catalytic activity and $H^+$ pumping activity after reconstitution into liposomes, and that the bulk proton gradient is fully dissipated in all proteoliposomes by the addition of $Ca^{2+}$ and PTP agonists, which rules out a minor contaminating protein as the agent of dissipation. The electrophysiological experiments indicate that permeabilization is due to opening of high-conductance channels that possess all the properties of the MMC/PTP, including (i) full reversibility of $Ca^{2+}$-dependent channel openings, (ii) sensitivity to well-characterized MMC/PTP activators and inhibitors in lipid bilayers, and (iii) conductance values and presence of substates of the measured channels[1,4]. Unique findings of the present work are the characterization of the single-channel activity elicited by $Ca^{2+}$ alone and the definition of the minimum $Ca^{2+}$ concentration required to trigger channel activity in vitro. Of note, no effects were elicited when the protein was added to the *cis* side and agonists to the *trans* side. Thus, the F-ATP synthase inserted with the hydrophylic $F_1$ sector facing the *cis* compartment; and the effect of agonists cannot be ascribed to unspecific effects on the lipid bilayer. Based on these results we see very little room for doubt that the permeabilization process in the liposomes and the currents measured in our experimental set-up are elicited by the bovine F-ATP synthase. We therefore conclude that F-ATP synthase is responsible for the formation of the MMC/PTP, as also strongly supported by our recent site-specific mutagenesis studies[11,12,26–28].

Native gel-eluted dimers and tetramers, but not monomers gave rise to currents after direct addition of the eluate to the *cis* side of the planar bilayer. The importance of this experiment lies in the fact that it further rules out the possibility that an unknown and difficult-to-detect contaminating protein may be

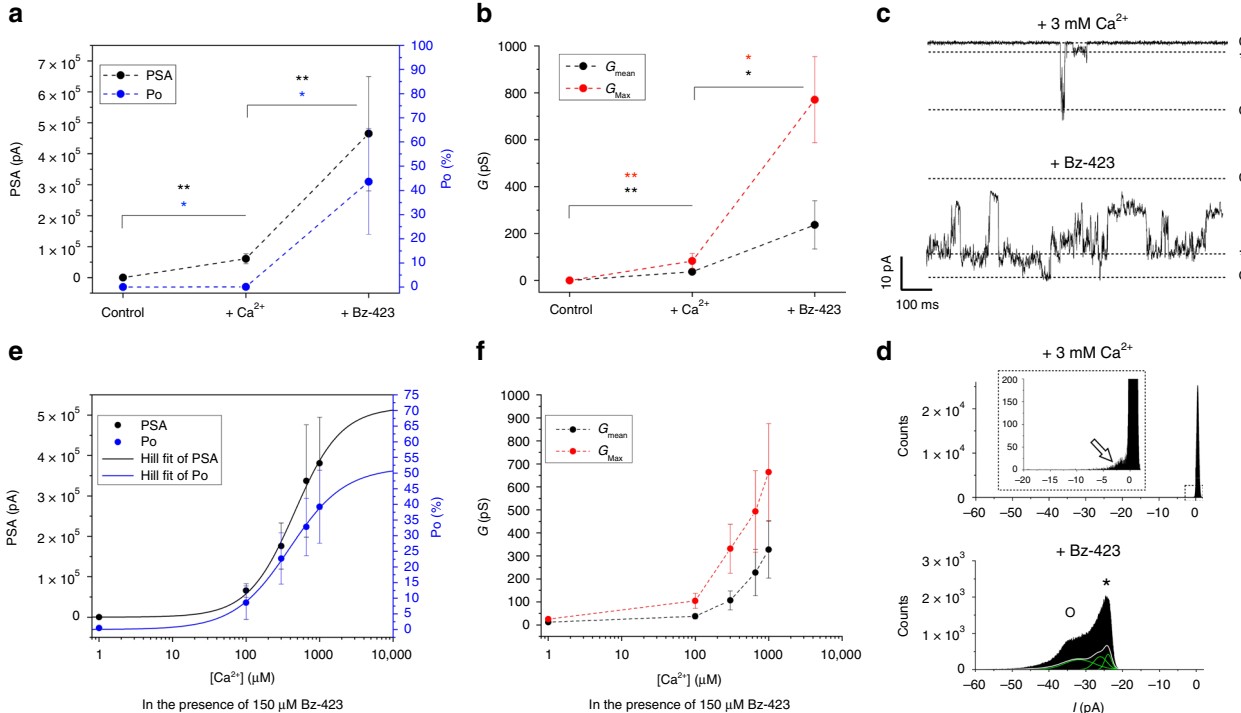

**Fig. 5** $Ca^{2+}$-dependence and activation of F-ATP synthase channels by Bz-423. **a** Power spectrum area (PSA) and open probability (Po) of F-ATP synthase channels were calculated from five independent experiments in the absence of activators (control), after addition of 3 mM $Ca^{2+}$ alone or following the addition of both 3 mM $Ca^{2+}$ and 0.15 mM Bz-423. Zero values are indicated with black and blue dotted lines for PSA and Po, respectively. Data shown are mean ± SEM. Statistical significance of differences was assessed with the Mann–Whitney test ($*p \le 0.05$, $**p \le 0.01$). **b** Mean and maximal conductance ($G_{mean}$ and $G_{max}$) were calculated for the same experiments shown in panel **a**. **c** Current traces elicited after addition of 3 mM $Ca^{2+}$ alone or after further addition of 0.15 mM Bz-423 ($V_{cis} = -60$ mV, $G_{max} = 500$ pS) from one representative experiment of five. The closed state is denoted with C, the most frequent subconductance state with an asterisk, and the maximal open state with O. **d** Current distribution of 30 s-recordings from experiments similar to the one reported in panel **c**. In the presence of $Ca^{2+}$ alone, flickering openings may be observed as a small peak in the histogram (arrow), while in the presence of Bz-423, a complex distribution with multiple peaks at higher conductance values (400 and 500 pS) is detectable. **e** PSA and Po of F-ATP synthase channels as a function of the $Ca^{2+}$ concentration in the presence of 0.15 mM Bz-423; PSA and Po distributions were fitted with the Hill equation ($n = 1.37 \pm 0.29$, $R^2 = 0.992$; $n = 1.17 \pm 0.16$, $R^2 = 0.997$, respectively). Data were obtained from 23 independent experiments and are shown as mean ± SEM. **f** $G_{mean}$ and $G_{max}$ of F-ATP synthase channels as a function of the $Ca^{2+}$ concentration from the same experiments shown in panel **e**. The source data underlying panels **a**, **b**, **e**, **f** are provided as a Source Data file

responsible for the recorded channel activity. Indeed, a putative contaminating channel of a size similar to F-ATP synthase might have co-migrated with the dimer or with the oligomer, but not with both; and a contaminating channel with high affinity for F-ATP synthase should have co-migrated with the monomer as well. Hence, the possibility that a protein different from F-ATP synthase is the underlying cause of the recorded currents becomes negligible. The absence of channel activity for the gel-eluted monomer and the equivalence of channel activity for the gel-eluted dimer and oligomer might suggest that the minimal channel-forming unit is the F-ATP synthase dimer. There are, however, several caveats to this interpretation. For example, the gel elution process might have stripped off factors essential for channel formation (such as structural lipids) from the monomers but not from the dimers and oligomers; or dimers reconstituted into the bilayer might have self-assembled into oligomers before the actual channel formation. Indeed, self-oligomerization of purified F-ATP synthase after lipid bilayer reconstitution has already been demonstrated for the yeast dimer by combining the mild GRecon membrane reconstitution approach with cryo-electron tomography of the resulting proteoliposomes[43,44]. Finally, channel formation itself might change the oligomeric state via monomerization. Thus, since we did not detect the oligomeric state and electric currents in the same experiment, the

conformation of F-ATP synthase during PTP formation remains an open question.

Oligomycin sensitivity and acidification of the internal volume indicate that close to all F-ATP synthase complexes incorporated in the liposomes exhibit their core function, i.e., coupled ATP hydrolysis and $H^+$ pumping; and characterization of the proteoliposomes clearly shows insertion of many F-ATP synthase complexes per liposome. Therefore, the detection of single-channel activity is surprising. A possible explanation is that only very few of the liposome-reconstituted F-ATP synthase complexes are MMC/PTP-competent, e.g., as a consequence of conformational heterogeneity. Alternatively, several complexes might participate in the measured currents with each single complex having only a low open probability, thus contributing to a few or even a single event during the total recording time. Irrespective of the underlying mechanism, however, it should be noted that the single-channel recordings reported here match the observation of MMC activity of patched mitoplasts[4], which should contain a high density of F-ATP synthase oligomers. Yet, single-channel activity can be observed also in the native membrane.

PTP formation has never been reported in prokaryotes. Mammalian oligomeric F-ATP synthase possesses several transmembrane subunits, which do not have homologs in the strictly monomeric chloroplast or bacterial F-ATP synthases[45,46], namely e, f, g, DAPIT, 6.8PL, and the newly found k subunit[20]. Therefore,

these are good candidates for playing a role in MMC/PTP formation. The recently published single particle cryo-EM structure of the porcine IF[1]-bound tetramer is the first structure of a mammalian F-ATP synthase where it is safe to assume that no subunits have been lost during the process of structure determination[20]. The structure revealed an intricate system of interactions between the monomers in the membrane-spanning F[O] domains, where the g–g and e–e interactions stand out as central. An unexpected finding of the structure is that the lumen of the c-ring seems to be filled by protein, putatively assigned to 6.8PL. This subunit makes a clear contact with the c-terminus of the long and bent α-helix of subunit e on the intermembrane space side of the complex. Both subunits e and g occupy a central role in oligomer formation[20]. Given the apparent conformational flexibility of subunit e, its prominent link to the c-ring lumen[47] and the crucial role of arginine 8 of subunit e for yeast F-ATP synthase MMC/PTP opening[28], it appears likely that subunit e is directly involved in MMC/PTP formation.

Formation of the MMC/PTP from F-ATP synthase has been questioned in recent studies of cells where the genes encoding for subunits c[48], b, and OSCP[49], and e, f, g, DAPIT, and 6.8PL[50] have been deleted. The key argument is that the PT persists in the absence of F-ATP synthase, given that deletion of any of these subunits prevents assembly of functional F-ATP synthase and of the respiratory chain[48–50]. We have addressed in detail the reasons why we think that the experiments of the first two publications are not conclusive[51,52]; and in our reading the most recent study[50] supports the conclusion that the PTP does originate from F-ATP synthase. Indeed, in wild-type cells $Ca^{2+}$-induced swelling in sucrose is immediate and complete in about 60 s, while mitochondrial swelling in cells lacking F-ATP synthase starts after a lag of about 30 s, proceeds at a very slow rate and is of limited extent[50], consistent with the lack of a bona fide PTP. We suspect that residual swelling activity and occurrence of $Ca^{2+}$-dependent $Ca^{2+}$ release in these cells is due to opening of the ANT channel[39,41], as suggested by its sensitivity to the selective ANT inhibitor BKA[50], which does not directly inhibit the MMC/PTP[53] nor the activity of the reconstituted F-ATP synthase channel (this manuscript).

While the main characteristics of the channel activity described here and those of MMC are in good agreement, differences also exist like the minimum $Ca^{2+}$ concentration needed to activate the MMC/PTP to full conductance, which is lower in mitoplasts from rat liver mitochondria[2,36]. This in turn suggests that additional factors—such as membrane curvature, lipid composition and interaction with other proteins—may contribute to regulation of the pore in intact, native membranes. Understanding the actual mechanism of pore formation will have to await experiments that are able to simultaneously detect the oligomeric state and channel opening, and the structural description of the open and closed channel states at the atomic level.

## Methods

**Purification of F-ATP synthase from bovine heart.** Purification of the lauryl-maltose-neopentyl glycol (LMNG) stabilized F-ATP synthase complex was conducted as previously described[15,30]. Briefly, fresh bovine hearts were obtained immediately after slaughter by an authorized slaughterhouse and inner mitochondrial membranes were purified according to Shinzawa-Itoh et al.[54] as follows. After careful removal of fat and connective tissues 1000 g of minced meat was prepared, and half of it was suspended in 3250 ml of 23 mM sodium phosphate buffer, pH 7.4, at 0 °C and homogenized for 5 min followed by centrifugation for 20 min at $2200 \times g$ in a large-scale refrigerated centrifuge (Kubota Model 9810) using an RS-6600 rotor. The other half of the minced meat was subjected to the same procedure. The combined precipitate was suspended in 3375 ml of 22.2 mM sodium phosphate buffer, pH 7.4, and re-homogenized, followed by centrifugation as before. All of the supernatants were combined and centrifuged for 30 min at $15,000 \times g$ with a refrigerated centrifuge (Beckman Model Avanti HP-30I) using a JLA-10.500 rotor. The precipitate, suspended in 50 mM Tris-HCl buffer, pH 8.0,

was centrifuged for 30 min at $75,000 \times g$ with an ultracentrifuge (Beckman Model-7) using a 45 Ti rotor. The precipitate was suspended in 50 mM Tris-HCl buffer, pH 8.0, containing 660 mM sucrose, adjusting the protein concentration to ~23 mg/ml. The suspension was kept in 40 mM HEPES pH 7.8, 2 mM MgCl[2], 0.1 mM EDTA, and 0.1 mM DTT and solubilized on ice via addition of deoxycholate and decylmaltoside to final concentrations of 0.7% (wt/vol) and 0.4% (wt/vol), respectively. Subsequently, the suspension was centrifuged at $176,000 \times g$ for 50 min and the supernatant applied to a sucrose step gradient (40 mM HEPES pH 7.8, 0.1 mM EDTA, 0.1 mM DTT, 0.2% wt/vol decylmaltoside and 2.0 M, 1.1 M, 1.0 M, or 0.9 M sucrose) and centrifuged at $176,000 \times g$ for 15.5 h. Fractions exhibiting ATPase activity determined by an ATP-regenerating enzyme-coupled assay[55] were loaded onto a Poros-20HQ ion-exchange column. The detergent was exchanged to LMNG using a double gradient from 0.2 to 0% decylmaltoside and 0%–0.05% LMNG for 80 min at 1 ml/min. Complexes were eluted by a linear concentration gradient of 0–240 mM KCl in 40 mM HEPES pH 7.8, 150 mM sucrose, 2 mM MgCl[2], 0.1 mM EDTA, 0.1 mM DTT, and 0.05% (wt/vol) LMNG. Shortly after elution, F-ATP synthase fractions containing high amounts of native phospholipids as determined by ammonium molybdate complexation were flash-frozen in aliquots of about 500 µl for later use.

**Lipid analysis of the purified F-ATP synthase preparation.** Conditions for the extraction of bovine F-ATP synthase were optimized to retain native phospholipids including cardiolipin. To this end, the total amount of phospholipids and relative amount of lipid species were monitored during purification as previously described[56]. The total amount of phospholipid contained in the F-ATP synthase preparation was determined by absorbance spectrometry (V-630- BIO spectrometer) at 600 nm using the phospholipid C-TestWako kit from Wako Fine Chemicals. The relative amount of native cardiolipin (CL), phosphoethanolamine (PE), and phosphocholine (PC) was analyzed by thin-layer chromatography. To extract the native lipids chloroform, methanol, and ammonia [65:2:33] were mixed with 10 mg of purified F-ATP synthase. After centrifugation at $2000 \times g$ for 15 min, the extracted lipids were removed and dried with N[2] gas. The lipid film was re-dissolved in 20 µl of chloroform, spotted onto a silica-gel plate (Merck, 60F254) and air-dried. The plate was developed with chloroform, methanol, ammonia, and water first at 6:4:0.3:0.2 and then at 75:22:3:0.1 ratios. The developed plate was completely dried and stained with 0.005% Primuline in 80% acetone. After air-drying, spots were scratched out and heated with Perchloric acid at 155 °C for 3 days to digest the extracted lipids completely. A 0.22% ammonium molybdate solution was added as Fiske-Subbarow reagent and mixed with digested lipids, and the solution was kept at 100 °C until onset of color change. Phosphorus content was determined by absorbance spectroscopy.

**Gel electrophoresis and protein elution from gels.** To confirm the subunit composition and intactness of the bovine F-ATP synthase after freezing and thawing for transport or for preparation of proteoliposomes, aliquots were subjected to denaturing sodium dodecyl sulfate polyacrylamide gel electrophoresis (SDS-PAGE) and non-denaturing clear-native PAGE[57]. For the experiments of Fig. 4c, 0.4 mg of purified F-ATP synthase were separated by BN-PAGE; monomers, dimers, and oligomers were identified, cut and eluted as described[7,58]. Briefly, bands were excised, diluted with 25 mM tricine, 7.5 mM Bis·Tris, 1% (w/v) n-heptyl β-D-thioglucopyranoside, pH 7.0, and supplemented with 8 mM ATP·Tris and 10 mM MgSO[4]. After overnight incubation at 4 °C, samples were centrifuged at $20,000 \times g$ for 10 min at 4 °C, and supernatants were used directly for reconstitution in electrophysiological studies.

**Negative staining electron microscopy.** To remove excess free LMNG an aliquot of 100 µl of purified bovine F-ATP synthase was subjected to a GraDeR run as described[30,59]. Subsequently a 2.5 µl aliquot was applied onto freshly glow-discharged, carbon-coated 400 mesh copper grids (Veco). After brief blotting (Whatman #1), the samples were stained by using a 2% uranyl acetate solution and air-dried. Images were taken with a JEM1010 transmission electron microscope (JEOL) equipped with a 4 x 4 K Tietz CMOS TemCamF416 (TVIPS, Gauting, Germany) at 100 kV and 20 pA/cm[2], an exposure time of 2 s and a magnification of $40,000 \times$, corresponding to a pixel size of 3.83 Å.

**(Proteo)liposome preparation.** Reconstitution was performed according to a published protocol[60]. Liposomes were prepared from purified soybean asolectin (L-α-phosphatidylcholine, Sigma). Lipids were dissolved in chloroform (5 mg/ml) until a homogeneous mixture was obtained, the solvent was then evaporated under a dry nitrogen stream to yield a thin layer of lipids at the bottom of a glass tube. The lipid film was thoroughly dried by placing the tube in a vacuum pump overnight to remove residual chloroform. Large multilamellar vesicles (LMV) were obtained by hydrating the lipid film with 1 ml of buffered solution (150 mM KCl, 10 mM Hepes, pH 7.4) and gently agitating at room temperature. Large unilamellar vesicles (LUV) were then prepared from LMV by extrusion through a polycarbonate filter with a pore size of 100 nm (extruder and membrane by Avanti Polar Lipids). F-ATP synthase or gramicidin was inserted into freshly made liposomes by direct incubation of the liposome solution for 30 min at 4 °C at a protein:asolectin ratio of 1:10 (w/w). After incubation, the solution was centrifuged

at 30,000 × g for 5 min, the supernatant removed, and fresh buffer added to resuspend the proteoliposomes. For biochemical assays, bovine heart lipids containing 1% ubiquinone wt/wt (Avanti Polar Lipids) in reconstitution buffer (50 mM MOPS pH 7.4, 30 mM NaCl, 100 mM KCl, 1.5 mM MgCl$_2$) were extruded 13 times through a 400 µM polycarbonate membrane at 38 °C. Liposomes were then incubated with the purified protein for 30 min at 25 °C with continuous mixing by inversion before use.

**Cryo-EM of liposome-reconstituted F-ATP synthase.** Pre-irradiated 200 mesh Quantifoil R2/2 molybdenum grids were glow-discharged and 3 µl of proteoliposome solution added to the grid. Excess solution was blotted with a GP Leica (Leica) followed by plunge freezing into liquid ethane after blotting for 10 s at 80% humidity and 20 °C. Data collection was performed on a JEM-3000SFF (JEOL) electron microscope at 300 kV with a field emission gun and a magnification of ×40,500 . The specimen stage temperature was maintained at ~80–100 K. The images were recorded on a K2 summit direct electron detector camera (Gatan) operated in a counting mode with a pixel size of 1.235 Å at the specimen level. Each image included 24 fractioned frames and image stacks were binned 2 × 2 by Fourier cropping, resulting in a pixel size of 2.47 Å. The stacked frames were subjected to motion correction with MotionCor2[61].

**Negative stain EM of liposome-reconstituted F-ATP synthase.** One drop of solution (about 25 µl) containing freshly prepared proteoliposomes was placed on 400 mesh holey film grid, stained with 1% uranyl acetate and observed with a Tecnai G2 (FEI) transmission electron microscope operating at 100 kV. Images were captured with a Veleta (Olympus Soft Imaging System) digital camera.

**ATPase activity in solution and after reconstitution.** ATP hydrolysis activity was measured at 38 °C with stirring at 1000 rpm using an ATP-regenerating assay[62]. Unless otherwise specified, the assay mixture contained 50 mM MOPS (pH 7.4), 30 mM NaCl, 100 mM KCl, 3 mM phospho*enol*pyruvate, 1.5 mM MgCl$_2$, 0.25 mM NADH, 0.57 U/ml pyruvate kinase, 3.2 U/ml lactate dehydrogenase, and 2 mM ATP. MgCl$_2$, CaCl$_2$, and ATP concentrations were varied as indicated in figure legends. The reaction was initiated by the addition of 10 µg of F-ATP synthase either in solution or after reconstitution into bovine heart lipids, into 1 ml of assay mixture. Where reconstituted enzyme was used, 1 µM valinomycin was included in the assay. The rate of NADH oxidation was monitored continuously at 340 nm using a modified Cary 60 spectrophotometer (Agilent). where indicated 2 µM oligomycin was added. The activity that hydrolyzed 1 µmol of ATP per min is defined as 1 unit.

**Determination of ATPase activity using Pi release.** ATP hydrolysis activity was measured in an assay containing 10 µg bovine F-ATP synthase, 50 mM MOPS (pH 7.4), 30 mM NaCl, and 100 mM KCl. When used, MgCl$_2$ and CaCl$_2$ concentrations were 2 mM, respectively. The reaction was initiated by the addition of sodium-ATP to a final concentration of 2 mM in a 1 ml of assay mixture. ATPase activity was measured at 38 °C with stirring at 1000 rpm using a colorimetric assay that detects inorganic phosphate liberated from ATP[63]. Nonenzymatic degradation of ATP under these conditions was <10% of the total phosphate. One unit of ATPase activity is defined as the amount of enzyme liberating 1 µmol of Pi or ADP/min at 38 °C.

**ATP-dependent proton translocation.** ATP-dependent proton translocation was determined at 38 °C based on the quenching of ACMA. The 1.5-ml reaction mixture contained 50 mM MOPS (pH 7.4), 30 mM NaCl, 100 mM KCl, 3 mM phospho*enol*pyruvate, 1.5 mM MgCl$_2$ 0.25 mM NADH, 0.57 U/ml pyruvate kinase, 3.2 U/ml lactate dehydrogenase, 1 µM ACMA, 1 µM valinomycin, and 10 µg F-ATP synthase complexes reconstituted in bovine heart lipids. Where indicated, 2.5 mM ATP was added. In some experiments, 0.125 mM Mg$^{2+}$ATP was used in a reaction mixture containing no MgCl$_2$, together with 30 µg F-ATP synthase complexes reconstituted in bovine heart lipids. After the fluorescence signal had stabilized, the reaction was initiated by the addition of the concentration of neutralized ATP indicated in the figure legends. Fluorescence was measured with an excitation wavelength of 410 nm and an emission wavelength of 480 nm (slit width, 10 nm) in a modified Cary Eclipse photospectrophotometer (Agilent).

**Protein digestion and liquid chromatography–mass spectrometry (LC-MS)/ MS analysis.** The preparation of F-ATP synthase complex obtained as above was processed using three different methods for protein digestion.

Two microliters of the F-ATP synthase complex preparation (8 µg/µl), were diluted with 18 µl of Laemmli sample buffer in reducing conditions (5% β-mercaptoethanol) and loaded onto a precast 4–12% SDS-PAGE slab (NuPAGE; Thermo Fisher Scientific). The electrophoretic process (30 mA constant) proceeded until bromophenol blue was completely inside the gel and proteins were focused in a single band. The gel was then stained with SimpyBlue Safe Stain (Invitrogen) and destained with water. The single band was manually excised, cut in small pieces and treated alternately with several washes of water and acetonitrile (ACN; Sigma). After the last wash with ACN, the gel pieces were dried under vacuum and then

treated with 200 µl of 10 mM DTT (Sigma) in 50 mM NH$_4$HCO$_3$ for 1 h at 56 °C and successively with 200 µl of 55 mM iodoacetamide (IAA; Sigma) in 50 mM NH$_4$HCO$_3$ for 45 min at room temperature and in the dark. The gel was then repeatedly washed with 200 µl of 50 mM NH$_4$HCO$_3$ and ACN. After dehydration under vacuum, gel pieces were incubated with 30 µl of sequencing-grade modified trypsin (12.5 ng/µl in 50 mM NH$_4$HCO$_3$; Promega) overnight at 4 °C. Peptides were extracted from the gel with three changes of 50% ACN/0.1% formic acid. The sample was dried under vacuum and kept at −20 °C until LC-MS/MS analysis.

Two microliters of the F-ATP synthase complex preparation (8 µg/µl) were diluted with 8 µl of 25 mM NH$_4$HCO$_3$, and treated with 10 µl of 10 mM DTT in 50 mM NH$_4$HCO$_3$ for 1 h at 56 °C. The sample was then alkylated with 20 µl of 55 mM IAA in 50 mM NH$_4$HCO$_3$ for 45 min at room temperature in the dark. Sequencing-grade modified trypsin (6.4 µl at 0.1 µg/µl in 50 mM NH$_4$HCO$_3$) was added to reach an enzyme to substrate ratio of 1:25. Digestion was carried on at 37 °C overnight, then 1 µl of pure formic acid (Sigma) was added to quench the reaction and the sample was kept at −20 °C until LC-MS/MS analysis was performed.

Five microliters of the F-ATP synthase complex preparation (8 µg/µl) were diluted with 195 µl of buffer A (urea 8 M, Tris-HCl 100 mM, pH 8.5) and subjected to a Filter-Aided Sample Preparation[64]. Briefly, the sample was loaded into a Vivacon 500 filter (10 kDa cutoff membrane; Sartorius) and centrifuged at 14,000 × g for 15 min. The flow-through was discarded and the process was repeated twice with 200 µl of buffer A. Ten microliters of 1 M DTT were added to 190 µl of buffer A and loaded into the filter. Incubation was carried out for 30 min at 55 °C, then the sample was centrifuged at 14,000 × g for 15 min and the flow-through was discarded. Ten microliters of 0.5 M IAA were added to 90 µl of buffer A and loaded into the filter. Alkylation proceeded for 30 min in the dark and at room temperature, then the sample was centrifuged at 14,000 × g for 15 min and the flow-through discarded. Two washing steps were performed with 100 µl of buffer A, followed by two additional washings with 100 µl of 100 mM NH$_4$HCO$_3$ and 50 mM NH$_4$HCO$_3$, respectively. Finally, 0.8 µg of sequencing-grade modified trypsin (12 ng/µl in 50 mM NH$_4$HCO$_3$) was added to the filter and digestion was carried out at 37 °C overnight. The sample was centrifuged at 14,000 × g for 15 min and the flow-through containing the digested proteins were retained, together with the subsequent washings performed using 100 µl of 50 mM NH$_4$HCO$_3$. One microliter of formic acid was added to the peptide mixture to quench the reaction, and the sample was diluted to a final volume of 1 ml using an aqueous solution of 0.1% formic acid. The mixture was finally desalted using a Sep-Pak C18 cartridge (Waters) following the manufacturer's instructions, dried under vacuum, and kept at −20 °C until LC-MS/MS analysis.

**LC-MS/MS and data analysis.** LC-MS/MS analysis was carried out using a LTQ-Orbitrap XL mass spectrometer (Thermo Fisher Scientific) coupled online to a nano-HPLC Ultimate 3000 (Dionex–Thermo Fisher Scientific). Samples were loaded into a NanoEase trap column Symmetry 300 (C18, 5 µm; Waters) at a flow rate of 8 µl/min using 0.1% formic acid and then transferred to a 10 cm pico-frit column (75 µm I.D., 15 µm tip; New Objective) packed in house with C18 material (Aeris Peptide 3.6 µm XB C18; Phenomenex). Peptides were separated using a linear gradient of ACN/0.1 formic acid from 3 to 40% in 40 min, using a flow rate of 250 nl/min. The instrument operated in data-dependent mode, with a Top4 acquisition method (one full scan in the 300–1700 m/z range at 60,000 resolution in the Orbitrap, followed by MS/MS of the four most intense ions acquired at low resolution in the linear ion trap). Ion source capillary temperature was set to 200 ° C, and spray voltage was set to 1.2−1.3 kV. Raw data files were processed with the software MaxQuant v.1.5.1.2[65] interfaced to the Andromeda search engine[66]. The search was done against the Bovine section of the Uniprot database (version 2017.07.26, 24333 entries) concatenated with a database of common contaminant found in proteomics studies. A parallel search against a reverse database was used to assess the quality of the data and to filter the results with a false-discovery rate (FDR) of 0.01, both at the peptide and protein level. Trypsin was set as enzyme with up to 2 missed cleavages allowed. Carbamidomethylation of Cys and oxidation of Met residues were set as fixed and variable modifications, respectively. The iBAQ parameter (intensity Based Absolute Quantification) was automatically calculated by the software and used as an estimate of the relative protein abundance level within sample[67]. Detailed results obtained by the analyses are reported in Supplementary Tables 1 and 2.

**Electrophysiology.** Electrophysiological properties of F-ATP synthase were assessed by means of single-channel recording following protein insertion into artificial planar lipid bilayers either from proteoliposomes (4 µg of protein), by direct addition of the purified protein (4 µg of protein) or from gel-purified proteins (1–2 µg of protein) as specified in text and legends. Proteoliposomes prepared as described or purified protein were added at a final concentration of 2.22 nM F-ATP synthase (based on the molecular mass of the monomer). For gel elution, 16 µg of purified protein were loaded in each lane followed by separation in BN-PAGE. Membranes were prepared by painting a solution of soybean asolectin (10 mg/ml in decane, Sigma) across a 250 µm-diameter hole on a teflon partition separating two compartments filled with a recording solution (150 mM KCl, 10 mM HEPES, pH 7.4) before membrane painting. The two compartments are identified as *cis* and *trans*, and all voltages refer to the *cis* side, zero being the *trans*

(grounded) one. Currents were considered as positive when carried by cations flowing from the *cis* to the *trans* compartment. Membrane capacity ranged from 50 to 150 pF (average 100 pF) and no current leakage was detectable. F-ATP synthase was added to the recording chamber (directly or as proteoliposomes as specified in the figure legends) followed by the additions specified in the text. Empty membranes did not display measurable currents during the whole recording time of up to 60 min ($n > 15$). Data were acquired at 10 kHz through a Bilayer Clamp BC-525C amplifier (Warner Instruments, Harvard Bioscience, Inc.) and low-pass filtered at 500 Hz. Data were digitized using a Digidata 1322 A interface and pClamp software (all from Molecular Devices) and analyzed offline using a home-developed algorithm described in Supplementary Tables 3 and 4 and based on MATLAB 2007b (MathWorks). The power spectrum of current signals was obtained using the Fast Fourier transform; frequencies lower than 2 Hz (baseline drifts), near 50 Hz (power line noise), and higher than 500 Hz (above the analog filter cutoff frequency) were not included in the analysis. The area under the power spectrum curve (referred as PSA) was calculated by integration in the whole selected frequency domain. The open probability (Po) of the channel was estimated as the ratio between the time spent in the open state (i.e., the time during which current amplitude was not in the noise range around 0 pA) and the whole recording time. Mean conductance ($G_{mean}$) was calculated from the average of the whole current signal in the open state on the basis of Ohm's law. Maximal conductance ($G_{max}$) was calculated on the basis of the maximal stable current level (i.e., events lasting at least 10 ms) in the recording interval. PSA, Po, $G_{mean}$, and $G_{max}$ were calculated for each experimental condition indicated in the text. In the voltage-dependence experiments data were corrected for the shift of about 13 pA caused by the imposed $Ca^{2+}$ gradient. Data are represented as mean ± SEM, and fits were calculated with Origin (OriginLab). PSA and Po distributions as a function of $Ca^{2+}$ concentration were fitted using the Hill equation[68]. Statistical comparison of data was assessed with the Mann–Whitney analysis.

**Protein assay**. Protein concentrations were determined using a bicinchoninic acid protein assay kit (Sigma) with bovine serum albumin as the standard.

These studies have complied with all relevant ethical regulations for animal testing and research.

**Reporting summary**. Further information on research design is available in the Nature Research Reporting Summary linked to this article.

## Data availability

Data supporting the findings of this manuscript are available from the corresponding authors upon reasonable request. A reporting summary for this Article is available as a Supplementary Information file. The mass spectrometry proteomics data have been deposited to the ProteomeXchange Consortium via the PRIDE[69] partner repository with the dataset identifier PXD015108. The source data underlying Fig. 1d, e, i, j, 2a, c–f, 3a, 4a, 5a, b, e, f and Supplementary Figs. 3a–d, 4, 8, and 11 are provided as a Source Data file.

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

## Acknowledgements

AIRC (grants IG17067 to P.B. and IG20286 to I.S.), Fondation Leducq (16CVD04 to P. B.), PRIN 2015795S5W and HFSP RG0052 (to I.S.), The Naito Foundation Subsidy for Female Researchers after Maternity Leave and JSPS 25-5370 to C.J.), CREST from JST (JPMJCR13M4 to Genji Kurisu and C.G.), Kiban B from MEXT (17H03647 to C.G.), Platform Project for Supporting Drug Discovery and Life Science Research [Basis for Supporting Innovative Drug Discovery and Life Science Research (BINDS)] from AMED (JP18am0101074) (to C.G. and K.A.), Delft Startup Grant (to D.G.G.M.). We wish to thank the "Cassa di Risparmio di Padova e Rovigo" Holding for funding the acquisition of the LTQ-Orbitrap XL mass spectrometer and Bernhard C. Ludewig for help with the graphics of Supplementary Fig. 1.

## Author contributions

Conceptualization (C.G., I.S., P.B.), Formal analysis (A.U., A.C., C.F., G.A.), Funding acquisition (C.G., I.S., P.B., C.J., K.A., D.G.G.M.), Investigation (A.U., V.G., A.C., C.F., C.J., K.A., S.M., K.S.-I., C.G., D.G.G.M.), Project administration (C.G., I.S., P.B.), Supervision (G.A., D.G.G.M., C.G., I.S., P.B.), Visualization (A.U., A.C., G.A., D.G.G.M., C.G.), Writing—original draft (C.G., I.S., P.B.), Writing—review and editing (D.G.G.M., C.G., I.S., P.B.).

## Competing interests

The authors declare no competing interests.
