## [Peer Review File · Nature Communications]

Reviewers' Comments:

Reviewer #1:

Remarks to the Author:

The authors have been characterizing further the potential role of the ATP synthase in forming the so-called "mitochondrial megachannel" that is activated upon, or is causative of, permeability transition. The authors are world leaders in this field and they are pursuing a very important area of study. This study is a follow up to their previous publication in which they purified ATP synthase by elution of dimers and monomers from BN-PAGE gels and reconstituted the dimers or monomers into planar lipid bilayers.

The present study has many more details regarding the ion channel activity and has an added set of recordings in which the highly purified ATP synthase is reconstituted into liposomes that are then fused with a planar lipid membrane for recordings.

The findings in this manuscript are similar to those in the previous study although the new method of purification is added. The authors themselves argue, however, that the channel recordings resulting from the two purification techniques are similar. Therefore it appears that the manuscript at this time offers an incremental advance over findings in the previous publication.

The channel recordings are very thorough in this study but there are questions remaining:

1. According to the cryo-EM image shown in Fig. 1f, reconstituted proteoliposomes are loaded with many ATP synthase molecules. How can the authors explain that they get single channel recordings in the experiment where they fuse these liposomes with the planar bilayers (recordings shown in Fig. 2)? How can they then explain the similar channel activity observed in recordings when they add the ATP synthase eluted from the BN-PAGE gel? This seems very unlikely given the enormous number of ATP synthases that are present in the liposome vs. the small amount of protein added directly to the bilayer. It seems the latter would give a higher probability of single channel activity.

2. How can the authors be sure that the dimers remain intact when they fuse either liposomes or eluted protein with the planar membrane? Would it not be the case that the angle for dimer formation would be incorrect in a planar membrane?

3. The authors should perform control experiments before using these fractions in planar bilayer recordings to assess whether the protein retains its oligomeric state after gel elution from BN-PAGE. They should also describe the elution procedure in detail: what was the detergent used to keep the protein solubilized after gel elution? How do they remove the detergent before using the protein for bilayer recordings?

4. The lipids used to form the liposomes and the lipid bilayer (as reported in the manuscript) do not contain cardiolipin. Cardiolipin has been suggested to be required for functional activity of ATP synthase dimers by maintaining the angle between the two monomers in the pair (Devrim Acehan,†† Ashim Malhotra,§ Yang Xu,§ Mindong Ren,† David L. Stokes,††¶ and Michael Schlame†§*Cardiolipin Affects the Supramolecular Organization of ATP Synthase in Mitochondria. *Biophys J.* 2011 May 4; 100(9): 2184–2192. doi: 10.1016/j.bpj.2011.03.031 PMID: PMC3150712; PMID: 21539786.

5. In all experiments, authors should report how many recordings resulted in no channel activity, since the amount of channel-forming protein added is very relevant to the study. For example: "7/8 recordings showed similar activity" or statements of that nature are essential. Was the amount of protein added to the planar lipid membrane the same in each case (liposomes, dimers, monomers, tetramers)?

6. In the authors' original papers on the permeability transition, they emphasized the finding that the PTP is activated upon mitochondrial membrane depolarization. Here they show mainly hyperpolarized potentials of -60 to -80 mV. Can the authors explain the discrepancy? They should show a graph of the likelihood of channel activity or the total or average current at the different holding voltages across the bilayer. Does calcium affect the channel differently at different voltages? Minor point: Recordings shown in Fig. 3c and in the extended data 5b have no information about holding voltages.

Reviewer #2:

Remarks to the Author:

The manuscript by Urbani and colleagues describes electrophysiological characterization of the ion channels formed by highly purified ATP synthase after its reconstitution into planar lipid bilayers. The key conclusion made by this study is that observed channel activity resembles that of native PTP channel and that due to the highly purified nature of the sample it is unlikely related to the other compounds that might be present in the prep. Further, authors conclude that such channel activity can only be observed in the preparations containing dimers and tetramers but not monomers of the ATP synthase. This is a highly significant work that provides solid evidence that ATP synthase complex presence is required for PTP activity. This work will be of great interest to the broad scientific audience. I have several specific issues that need to be addressed:

1. As argued in the manuscript, ATP synthase can't actually form channels but requires assembly into dimers and tetramers. This makes title and abstract (as they presented now) somewhat confusing and contradictory to what manuscript shows. These need to be modified.
2. There is certain non-trivial relationship between EM and bilayer data that needs to be carefully considered and discussed. EM images clearly show presence of multiple complexes of the ATP synthase per liposome, which is also confirmed by ATPase activity. This implies that when these vesicles are fused with planar bilayer multiple complexes will get incorporated. It needs to be explained why addition of calcium doesn't induce multiple channels opening but rather single channel activity? This suggests that very small fraction of ATP dimers can actually be transformed into PTP by calcium. In other words, ATP dimers (at least in their native conformation) are essential but, apparently not sufficient for PTP formation. What is it: presence of some "rare/essential" binding partner? some unorthodox conformation state? ... missing "channel part" associated with small fraction of dimers?
3. It would strengthen the paper if authors tested activation by recombinant/purified CypD (followed by CSA block) of the channel activity. Taking into account that CSA is considered to be the defining feature of PTP this experiment would strongly solidify presented evidence.

Other suggestions:

1. Authors should provide more details of PSA analysis by including references and perhaps algorithm into the supplemental data.
2. Overall bilayer data need to be carefully reviewed and important details added and corrected. For example, please define how long did it take to channel to activate (or be inhibited) after calcium(drug) application. As presented now reader can only see "before/after" traces. Further, authors should carefully clarify how variable the channel activity was for experiment to experiment. For example, Fig. 2A shows considerable activation by 3 mM calcium while Fig. 3C shows virtually no activity at the same calcium concentration. It would help if authors plotted complete Ca data set similar to what they have now for Ca+Bz (Ext data Fig. 8). Labels need to be checked carefully throughout, for example ED Fig 7 top panel "o" state has multiple counts (as judged by histogram) but few events if judged by the trace.
3. Have authors tried to add Ca at the end to the experiment (Fig. 1j)? This seems to be logical control to confirm loss of membrane integrity to make sure that detected conductance is not only relevant to the bilayer techniques.
4. In addition to MMC authors should mention another term that has been coined for PTP-like activity – "MCC" or "Multiple Conductance Channel". This is fair from historical perspective and relevant to the behavior they see in the bilayers.
5. Was there any statistical difference in channel activity depending on the reconstitution method (liposomes vs gel-purified)?

Reviewer #3:

Remarks to the Author:

An international team of ATP synthase experts has come together to investigate the long-standing question about the physical nature of the mitochondrial permeability transition pore (mPTP). For

many decades this has been a highly controversial topic; it was discussed in many papers, reviews and conferences. Today, the prime suspects of the mPTP, causing membrane depolarisation and cell death (apoptosis), are the mitochondrial ATP synthase, the adenine nucleotide translocase (ANT) and the voltage-dependent anion channel (VDAC). For this work, the bovine mitochondrial ATP synthase purified using the mild detergent LMNG. The material presented in Fig. 1 looks excellent, is highly pure and active (but the specific activities, Fig. 1d, should be calculated and provided). The inhibition by oligomycin as well as ACMA quenching experiments, showing coupling and proton transport, has been carefully done and looks healthy. The authors then continued to investigate ion conductance properties of the proteoliposome-reconstituted ATP synthase with the specific aim to look at its capability to conduct Ca^{2+} . Fig. 2 shows the channel activities, with control experiments. The electrophysiological trace shown in Fig. 2f is the key experiment of this work: from the cis side (F1 oriented) the ATP synthase is capable of forming a conductive pore/channel/pathway for Ca^{2+} , in the presence of the physiological channel modulator Bz-423 (cyclophilin D mimic, induces mPTP opening in mitochondria). Furthermore, the experiment shown in Fig. 2g demonstrates channel activity of isolated ATP synthase dimers and higher oligomers but, notably, not with the isolated, monomeric form of ATP synthase. This experiment is equally important and can perhaps help explain past controversial experiments or findings. Due to the importance of this new information this key insight should also be anchored in the abstract, while I do see less relevance of having the Power Spectrum Area (PSA) experiment shown in the main section (or maybe I did not really get the point). Finally, the authors also present convincing data to exclude the possibility that potential (very minor traces, if at all) of ANT and/or VDAC, or any other putative contaminating compounds in the preparation might play a role in the effects shown in these experiments. I was very impressed how deep and careful all these possibilities were investigated in this work.

The manuscript is concise and brings the problem to the point. It should stay like this. The data is clear, and the conclusions are justified. The work is highly important contribution for the fields of bioenergetics and cell biology. I highly recommend publication.

Some points:

The two subunits DAPIT and 6.8PL seem to be coming in the focus being involved in forming the mPTP, perhaps also the dimer supporting subunits e and g, or subunit f that has not yet been excluded (ref. 30, He et al, PNAS 2017). This could be shortly mentioned in the discussion. Some more information about these subunits (e.g. their location in the structure) would help. On page 2, second paragraph: subunit 6.8 proteolipid was found but it is not in Extended Data Table 2. Please clarify (nomenclature...).

The authors state on page 3 that the cyclophilin D mimic Bz-423 binds to OSCP, in agreement with Johnson et al (2005) Chem. Biol. 12: 485, but newer data (ref. 30) strongly argues against cyclophilin D is capable of binding to OSCP. Maybe it has an alternative binding site in ATP synthase? This point will remain open for future clarification and the statements in this work should be modified accordingly.

Dear Dr. Bernardi,

Your manuscript entitled "The mitochondrial permeability transition pore is a Ca^{2+} -dependent channel formed by F-ATP synthase" has now been seen by 3 referees. You will see from their comments below that while they find your work of interest, some important points are raised. We are interested in the possibility of publishing your study in Nature Communications, but would like to consider your response to these concerns in the form of a revised manuscript before we make a final decision on publication.

Thank you very much for the opportunity to resubmit. We have rarely found the Reviewers' comments to be so useful, and we have carefully addressed all the points raised. As a result, several Extended Data figures and tables have been added, including a "flow chart" that was not requested but that we hope you will like; the title of the manuscript has been changed to reflect the centrality of F-ATP synthase in the study; and the reorganization as per Nature Communications style led to an expansion of the discussion. We feel that the revised manuscript has been improved substantially, and we are very grateful for this great opportunity to strengthen our observations. The point-by-point reply can be found below.

Reviewers' comments:

Reviewer #1 (Remarks to the Author):

The authors have been characterizing further the potential role of the ATP synthase in forming the so-called "mitochondrial megachannel" that is activated upon, or is causative of, permeability transition. The authors are world leaders in this field and they are pursuing a very important area of study. This study is a follow up to their previous publication in which they purified ATP synthase by elution of dimers and monomers from BNP gels and reconstituted the dimers or monomers into planar lipid bilayers.

The present study has many more details regarding the ion channel activity and has an added set of recordings in which the highly purified ATP synthase is reconstituted into liposomes that are then fused with a planar lipid membrane for recordings.

The findings in this manuscript are similar to those in the previous study although the new method of purification is added. The authors themselves argue, however, that the channel recordings resulting from the two purification techniques are similar. Therefore it appears that the manuscript at this time offers an incremental advance over findings in the previous publication.

The channel recordings are very thorough in this study but there are questions remaining:

1. According to the cryo-EM image shown in Fig. 1f, reconstituted proteoliposomes are loaded with many ATP synthase molecules. How can the authors explain that they get single channel recordings in the experiment where they fuse these liposomes with the planar bilayers (recordings shown in Fig. 2)? How can they then explain the similar channel activity observed in recordings when they add the ATP synthase eluted from the BN-PAGE gel? This seems very unlikely given the enormous number of ATP synthases that are present in the liposome vs. the small amount of protein added directly to the bilayer. It seems the latter would give a higher probability of single channel activity.

Answer. This comment (also made by the second Reviewer) has been extremely useful. We have realized that not enough attention had been paid to address and explain this point. We have assessed the distribution of F-ATP synthase in the liposomes, and found that it associates with small liposomes, which are known to have a low probability of fusion with planar membranes particularly when an osmotic gradient is not applied. We have also carried out parallel experiments using gramicidin as the test channel, and confirmed that incorporation is a rare event in our experimental conditions. The proteoliposome approach was important as a proof of principle, as we also show that the F-ATP synthase is fully active as a proton pump; but indeed most experiments have been performed by direct addition of the purified preparation to the bilayer. All of the above has been addressed in the revision with a modified Fig. 1 (which

now contains the distribution of F-ATP synthase according to liposome size), a new figure with gramicidin (Extended Data Fig. 6), and a thorough discussion with appropriate references.

2. How can the authors be sure that the dimers remain intact when they fuse either liposomes or eluted protein with the planar membrane? Would it not be the case that the angle for dimer formation would be incorrect in a planar membrane?

Answer. To address this problem we have shown by resolubilization of proteliposomes with digitonin followed by BN-PAGE analysis that a substantial fraction of F-ATP synthase remains in the dimeric/oligomeric form (Extended Data Fig. 5). We cannot be sure of the oligomeric state of the protein in the bilayer, although it appears that starting from the gel-purified monomers channels do not form. Regarding the question about the angle for dimer formation in the planar membrane, we would like to point out that the mitochondrial ATP synthase induces local curvatures in lipid bilayers due to its molecular shape. This has been demonstrated by cryo-EM tomography on proteoliposomes containing both monomeric and dimeric F-ATP synthase. Interestingly, self-assembly of dimers into oligomeric rows has been demonstrated with the same approach, see Blum et al., Dimers of mitochondrial ATP synthase induce membrane curvature and self-assemble into rows. *Proc. Natl. Acad. Sci. U. S. A.* **116**, 4250-4255 (2019) and Jiko, et al., Bovine F1Fo ATP synthase monomers bend the lipid bilayer in 2D membrane crystals. *eLife Sciences* **4**, e06119 (2015). This important point has now been discussed and referenced.

3. The authors should perform control experiments before using these fractions in planar bilayer recordings to assess whether the protein retains its oligomeric state after gel elution from BN-PAGE. They should also describe the elution procedure in detail: what was the detergent used to keep the protein solubilized after gel elution? How do they remove the detergent before using the protein for bilayer recordings?

Answer. We have added a paragraph with the details in Methods. The elution procedure was developed by Rehling et al. [Protein insertion into the mitochondrial inner membrane by a twin-pore translocase. *Science* **299**, 1747-1751 (2003)] for reconstitution of the twin pore translocase, and the buffer contained 1% (w/v) n-heptyl β -D-thioglucoopyranoside. The eluted dimers retained at least in part the dimeric structure, as we showed in Giorgio et al. (2013). Unlike the original study, however, the point of the elution experiment is not to make claims on the oligomeric state of the "active" F-ATP synthase (which remains unknown), but to further address the possibility of an unknown trace contaminant causing the recorded currents.

4. The lipids used to form the liposomes and the lipid bilayer (as reported in the manuscript) do not contain cardiolipin. Cardiolipin has been suggested to be required for functional activity of ATP synthase dimers by maintaining the angle between the two monomers in the pair (Devrim Acehan,†† Ashim Malhotra,§ Yang Xu,§ Mindong Ren,† David L. Stokes,††¶ and Michael Schlame†§*Cardiolipin Affects the Supramolecular Organization of ATP Synthase in Mitochondria. *Biophys J.* 2011 May 4; 100(9): 2184–2192. doi: 10.1016/j.bpj.2011.03.031 PMID: 21539786.

Answer. Thank you very much for pointing out this important aspect of F-ATP synthase preparation from native tissue. Indeed cardiolipin is essential for the proper function of F-ATP synthase and other multisubunit complexes of the inner mitochondrial membrane. We have observed in our purification experiments that depletion of native cardiolipin during the purification, especially at the Poros column step, destabilizes the complex. This is in agreement with the finding that cardiolipin binds tightly to F-ATP synthase and likely plays a structural role [Eble et al. (1990) Tightly associated cardiolipin in the bovine heart mitochondrial ATP synthase as analyzed by ³¹P nuclear magnetic resonance spectroscopy. *J. Biol. Chem.* **265**, 19434–19440]. Therefore, we have optimized our preparation method to retain more than 100 phospholipid molecules per F-ATP synthase and a cardiolipin ratio that is not below that found in the inner mitochondrial membrane. In our hands this was crucial for obtaining stable, tightly coupled and well behaving complexes suitable for structural and functional studies [see Jiko et al., Bovine F1Fo ATP synthase monomers bend the lipid bilayer in 2D membrane crystals. *eLife Sciences* **4**, e06119 (2015) and Hauer et al. GraDeR: Membrane Protein Complex Preparation for Single-Particle Cryo-EM. *Structure* **23**, 1769-1775

(2015)]. In order to properly convey this aspect of our study to the reader, we have now included information on content of cardiolipin and other lipids in the main text and added a new specific Methods Section on lipid analysis.

5. In all experiments, authors should report how many recordings resulted in no channel activity, since the amount of channel-forming protein added is very relevant to the study. For example: “7/8 recordings showed similar activity” or statements of that nature are essential. Was the amount of protein added to the planar lipid membrane the same in each case (liposomes, dimers, monomers, tetramers)?

Answer. We observed activity as follows: Proteoliposomes, 13 out of 16 (81%); direct reconstitution, 103 out of 131 (79%), which compares well with gramicidin, 6 out of 8 (75%); gel-eluted proteins, 9 out of 10 for oligomers (90%), 8 out of 9 for dimers (89%) and 0 out of 7 (0%) for monomers. This is now specified in the main text. The amount of protein added was 4 μg for proteoliposomes and purified protein, and 1-2 μg for the eluted bands. This is now specified in the Methods Section and in the text, thank you for pointing out these important issues.

6. In the authors' original papers on the permeability transition, they emphasized the finding that the PTP is activated upon mitochondrial membrane depolarization. Here they show mainly hyperpolarized potentials of -60 to -80 mV. Can the authors explain the discrepancy? They should show a graph of the likelihood of channel activity or the total or average current at the different holding voltages across the bilayer. Does calcium affect the channel differently at different voltages? Minor point: Recordings shown in Fig. 3c and in the extended data 5b have no information about holding voltages.

Answer. In patch-clamp experiments the voltage is lower than that maintained by mitochondria (which is about 180 mV, negative inside) because the physiological voltage would cause breakdown of the bilayer. Thus, all these experiments are actually performed at voltages that would favor pore opening in mitochondria. We have performed a study of the I/V relationship, which is reported in panel a of new Figure 4. We have studied the effect of Ca^{2+} on channel activity, while we would like to defer a study of the effects on the voltage-dependence to a specific study due to complexity of the experiments and to the number of replicates involved. Holding voltage is now specified.

Reviewer #2 (Remarks to the Author):

The manuscript by Urbani and colleagues describes electrophysiological characterization of the ion channels formed by highly purified ATP synthase after its reconstitution into planar lipid bilayers. The key conclusion made by this study is that observed channel activity resembles that of native PTP channel and that due to the highly purified nature of the sample it is unlikely related to the other compounds that might be present in the prep. Further, authors conclude that such channel activity can only be observed in the preparations containing dimers and tetramers but not monomers of the ATP synthase. This is a highly significant work that provides solid evidence that ATP synthase complex presence is required for PTP activity. This work will be of great interest to the broad scientific audience.

Answer. Thank you for the very supportive comments.

I have several specific issues that need to be addressed:

1. As argued in the manuscript, ATP synthase can't actually form channels but requires assembly into dimers and tetramers. This makes title and abstract (as they presented now) somewhat confusing and contradictory to what manuscript shows. These need to be modified.

Answer. Thank you for the comment, which led to a change of the title to “Purified F-ATP synthase forms a Ca^{2+} -dependent high-conductance channel matching the mitochondrial permeability transition pore”,

which now puts the F-ATP synthase at center stage. We have also discussed more thoroughly the potential relevance (but also the limits) of the gel elution experiments.

2. There is certain non-trivial relationship between EM and bilayer data that needs to be carefully considered and discussed. EM images clearly show presence of multiple complexes of the ATP synthase per liposome, which is also confirmed by ATPase activity. This implies that when these vesicles are fused with planar bilayer multiple complexes will get incorporated. It needs to be explained why addition of calcium doesn't induce multiple channels opening but rather single channel activity? This suggests that very small fraction of ATP dimers can actually be transformed into PTP by calcium. In other words, ATP dimers (at least in their native conformation) are essential but, apparently not sufficient for PTP formation. What is it: presence of some "rare/essential" binding partner? some unorthodox conformation state? ... missing "channel part" associated with small fraction of dimers?

Answer. You are absolutely right, and this point (also raised by the first Reviewer) has been extremely useful. We have realized that we did not pay enough attention to address and explain it. We have assessed the distribution of F-ATP synthase in the liposomes, and found that it associates with small liposomes, which are known to have a low probability of fusion with planar membranes particularly when an osmotic gradient is not applied. We have also carried out parallel experiments using gramicidin D as the test channel, and confirmed that incorporation is a rare event in our experimental conditions. The proteoliposome approach was important as a proof of principle, as we also show that the F-ATP synthase is fully active as a proton pump; but indeed most experiments have been performed with direct addition of the purified preparation. All of the above (including statistics of the "success rate" of each procedure, please also see our reply to Reviewer 1) has been addressed in the revision with a modified Fig. 1 (which now contains the size distribution), a new figure with gramicidin (Extended Data Fig. 6), and a thorough discussion with appropriate references.

3. It would strengthen the paper if authors tested activation by recombinant/purified CypD (followed by CSA block) of the channel activity. Taking into account that CSA is considered to be the defining feature of PTP this experiment would strongly solidify presented evidence.

Answer. We have used Bz-423 because it is a *bona fide* substitute of CyPD, and we preferred it because of availability, stability and lack of interference with the experimental setup (this is now specified in the main text). We agree that using CyPD would be a plus, but given the number of experiments we had to perform in the revision and the limited amount of time for resubmission we have deferred these experiments. This is the only request that we have not fulfilled, but in recognition of its importance we have set up the production of the recombinant protein and we plan to perform these experiments in the follow-up of the present studies.

Other suggestions:

1. Authors should provide more details of PSA analysis by including references and perhaps algorithm into the supplemental data.

Answer. The algorithms has been provided in new Extended Data Tables 4 and 5.

2. Overall bilayer data need to be carefully reviewed and important details added and corrected. For example, please define how long did it take to channel to activate (or be inhibited) after calcium(drug) application. As presented now reader can only see "before/after" traces.

Answer. Thank you, we have realized that this needed to be addressed and we are happy that we had a chance to revise the manuscript. This point (including also statistics of the "success rate" of each procedure, please also see our reply to Reviewer 1) has been addressed in the revision and the information can now be found in the text and in new Extended Data Fig. 8.

Further, authors should carefully clarify how variable the channel activity was for experiment to experiment. For example, Fig. 2A shows considerable activation by 3 mM calcium while Fig. 3C shows virtually no activity at the same calcium concentration. It would help if authors plotted complete Ca data set similar to what they have now for Ca+Bz (Ext data Fig. 8). Labels need to be checked carefully throughout, for example ED Fig 7 top panel “o” state has multiple counts (as judged by histogram) but few events if judged by the trace.

Answer. Please note that the figure numbering has changed in the revised manuscript, and that we are referring to the present version. Thank you very much for pointing this out. As also mentioned in the text, currents observed with Ca^{2+} alone are smaller (Fig. 3a) than those observed in the presence of Bz-423 (Fig. 3b, please note that the ordinate scale (pA) is less expanded than in 3a). In order to highlight the effect of Bz-423, in Fig. 5c we had selected a 1s-current trace where openings weren't frequent (although some flickering was visible). Currents induced by Ca^{2+} alone were detected (they were indicated by the arrow in the histogram, which covers a recording lasting 30s). We now realize that this was confusing, and in Fig. 5c we have now used a 1s-current trace from the same recording where some openings induced by Ca^{2+} alone can be appreciated. To further clarify this point we have also added a new Extended Data Figure 13.

3. Have authors tried to add Ca at the end to the experiment (Fig. 1j)? This seems to be logical control to confirm loss of membrane integrity to make sure that detected conductance is not only relevant to the bilayer techniques.

Answer. This comment generated a new set of experiments on the proteoliposomes, which are reported in new Fig. 2. We do find rapid and complete dissipation of the H^+ gradient by Ca^{2+} in the presence of MMC/PTP stabilizers. This is a key finding that considerably strengthens the manuscript because it shows for the first time a bulk solution measurement of MMC/PTP-like behaviour by all F-ATP synthase-acidified proteoliposomes in an *in vitro* assay. This rules out any possibility of a contaminant being responsible for the observed permeabilization. We feel that this is a key addition to the manuscript and we are very grateful for the suggestion.

4. In addition to MMC authors should mention another term that has been coined for PTP-like activity – “MCC” or “Multiple Conductance Channel”. This is fair from historical perspective and relevant to the behavior they see in the bilayers.

Answer. Done, and proper reference added.

5. Was there any statistical difference in channel activity depending on the reconstitution method (liposomes vs gel-purified)?

Answer. In the liposome experiment we only used Ca^{2+} to activate the channel, which as discussed above generates smaller currents, so this reconstitution condition cannot be compared to the direct addition and gel elution methods.

Reviewer #3 (Remarks to the Author):

An international team of ATP synthase experts has come together to investigate the long-standing question about the physical nature of the mitochondrial permeability transition pore (mPTP). For many decades this has been a highly controversial topic; it was discussed in many papers, reviews and conferences. Today, the prime suspects of the mPTP, causing membrane depolarisation and cell death (apoptosis), are the mitochondrial ATP synthase, the adenine nucleotide translocase (ANT) and the voltage-dependent anion channel (VDAC). For this work, the bovine mitochondrial ATP synthase purified using the mild detergent LMNG. The material presented in Fig. 1 looks excellent, is highly pure and active (but the specific activities, Fig. 1d, should be calculated and provided). The inhibition by oligomycin as well as ACMA quenching

experiments, showing coupling and proton transport, has been carefully done and looks healthy. The authors then continued to investigate ion conductance properties of the proteoliposome-reconstituted ATP synthase with the specific aim to look at its capability to conduct Ca^{2+} . Fig. 2 shows the channel activities, with control experiments. The electrophysiological trace shown in Fig. 2f is the key experiment of this work: from the cis side (F1 oriented) the ATP synthase is capable of forming a conductive pore/channel/pathway for Ca^{2+} , in the presence of the physiological channel modulator Bz-423 (cyclophilin D mimic, induces mPTP opening in mitochondria). Furthermore, the experiment shown in Fig. 2g demonstrates channel activity of isolated ATP synthase dimers and higher oligomers but, notably, not with the isolated, monomeric form of ATP synthase. This experiment is equally important and can perhaps help explain past controversial experiments or findings.

Answer. Thank you very much for the very positive and encouraging comments.

Due to the importance of this new information this key insight should also be anchored in the abstract,

Answer. As suggested, we have included this finding in the abstract.

while I do see less relevance of having the Power Spectrum Area (PSA) experiment shown in the main section (or maybe I did not really get the point).

Answer. We decided to keep this experiment in the main text because it provides quantitative information on the Ca^{2+} -dependence of the F-ATP synthase channel, and this is done for the first time.

Finally, the authors also present convincing data to exclude the possibility that potential (very minor traces, if at all) of ANT and/or VDAC, or any other putative contaminating compounds in the preparation might play a role in the effects shown in these experiments. I was very impressed how deep and careful all these possibilities were investigated in this work.

Answer. We are truly grateful for the appreciation of our efforts.

The manuscript is concise and brings the problem to the point. It should stay like this. The data is clear, and the conclusions are justified. The work is highly important contribution for the fields of bioenergetics and cell biology. I highly recommend publication.

Answer. Thank you, we had to expand the manuscript with a Discussion Section to comply with the format of Nature Communications, but hope that the manuscript is still as effective.

Some points:

The two subunits DAPIT and 6.8PL seem to be coming in the focus being involved in forming the mPTP, perhaps also the dimer supporting subunits e and g, or subunit f that has not yet been excluded (ref. 30, He et al, PNAS 2017). This could be shortly mentioned in the discussion. Some more information about these subunits (e.g. their location in the structure) would help.

Answer. Though our study does not address the question of which subunits are involved in the actual channel formation, this of course a very important and interesting and still unresolved point. We have now included a brief discussion of likely candidates as follows (references can be found in the Discussion):

"PTP formation has never been reported in prokaryotes. Mammalian oligomeric F-ATP synthase possesses several transmembrane subunits which do not have homologues in the strictly monomeric chloroplast or bacterial F-ATP synthases, namely e, f, g, DAPIT, 6.8PL and the newly found k subunit. Therefore, these are good candidates for playing a role in MMC/PTP formation. The recently published single particle cryo-EM structure of the porcine IF_1 -bound tetramer is the first structure of a mammalian F-ATP synthase where it is

safe to assume that no subunits have been lost during the process of structure determination. The structure revealed an intricate system of interactions between the monomers in the membrane-spanning F_0 domains, and the g-g and e-e interactions stand out as central. An unexpected finding of the structure is that the lumen of the c-ring seems to be filled by protein, putatively assigned to 6.8PL. This subunit makes a clear contact with the c-terminus of the long and bent α helix of subunit e on the intermembrane space side of the complex. Both subunits e and g occupy a central role in oligomer formation. Given the apparent conformational flexibility of subunit e, its prominent link to the c-ring lumen and the crucial role of arginine 8 of subunit e for yeast F-ATP synthase MMC/PTP opening, it appears likely that subunit e is directly involved in MMC/PTP formation."

On page 2, second paragraph: subunit 6.8 proteolipid was found but it is not in Extended Data Table 2. Please clarify (nomenclature...).

Answer. 6.8PL was detected by MALDI-TOF MS performed specifically to detect smaller MW subunits as reported in Extended Data Table 1. This subunit was also repeatedly detected by LC-MS/MS in samples extracted from BN gels. However, it was not detected by quantitative analysis of the protein composition of the F-ATP synthase preparation (as now mentioned in Extended Data Table 2) which was specifically tailored to detect contaminating proteins.

The authors state on page 3 that the cyclophilin D mimic Bz-423 binds to OSCP, in agreement with Johnson et al (2005) Chem. Biol. 12:485, but newer data (ref. 30) strongly argues against cyclophilin D is capable of binding to OSCP. Maybe it has an alternative binding site in ATP synthase? This point will remain open for future clarification and the statements in this work should be modified accordingly.

Answer. The data of He et al. do not argue against CyPD binding to OSCP, they rather suggest that the PTP is sensitive to CsA even in the absence of OSCP, implying that OSCP is not the CsA-sensitive binding site for CyPD as a PTP inhibitor. We think that John Walker is misinterpreting his results, and we have avoided any polemics by referring the readers to already published comments. However, we do comment on his latest PNAS, which appeared on June 18, and we discuss new data suggesting a second possibility, i.e. that there is a second CsA-sensitive and cyclophilin D-stimulated channel (most likely the ANT). This was recently shown by Neginskaya et al. in the first independent assessment of the sub c null cells generated by the Walker lab. The issue is now addressed in the discussion, and we hope that our account is fair and useful.

Reviewer 4 (late)

Review of **"The Mitochondrial Permeability Transition Pore is a Ca^{2+} -Dependent Channel Formed by F-ATP Synthase"**

By A Urbani, V Giorgio, A Carrer, C Franchin, G Arrigoni, C Jiko, K Abe, J FM Bogers, DGG McMillan, C Gerle, I Szabò, and P Bernardi

Synopsis:

Using highly purified preparations of the bovine F-ATP synthase, this paper reports that dimers and tetramers, but not monomers, of the F-ATP synthase can form Ca^{2+} plus benzodiazepine-423 (Bz-423) activated ion channels. Moreover, these F-ATP synthase channels have a number of features in common with those of the mitochondrial permeability transition pore (PTP).

The bovine F-ATP synthase was purified from bovine heart by sucrose gradient and ion exchange chromatography using the detergent lauryl-maltose-neopentyl glycol (LMNG). Purity was assessed by blue native gel electrophoresis, SDS-polyacrylamide gel electrophoresis, and mass spectrometry. All subunits could be identified (Figure 1c). Analysis by mass spectrometry showed that the major subunits could be detected, though the α and β subunits were not in the customary 3:1 stoichiometry relative to the central axial subunits (Extended Data Table 2 and Fig. 1)).

The mass spectrometry data permitted the determination of other proteins previously linked to the PTP in the preparation. This revealed both ADP/ATP translocase 1 and voltage dependent anion-selective channel (VDAC) contamination but at low levels. The ADP/ATP translocator 1 was only 1.26% of the α subunit, 1.9%

of the β subunit, and 6.8% of the b subunit. The VDAC proteins 1 & 2 were but half the abundance of the ADP/ATP translocase 1. Thus, the F-ATP synthase used in these studies was quite pure.

The purified F-ATP synthase was incorporated into liposomes and the proteoliposomes were used to demonstrate that the purified F-ATP synthase retained the classical functions of the F-ATP synthase. The F-ATP synthase containing proteoliposomes were then fused to a lipid film separating *cis* and *trans* chambers and the F-ATP synthase containing films used to study the channel forming characteristics of the F-ATP synthase. The F-ATP synthase containing lipid films were then shown to form ion permeable channels when exposed to Ca^{2+} with the amount of current increasing with the Ca^{2+} concentration. The capacity of the F-ATP synthase to form the Ca^{2+} ion channels was enhanced by Bz-423 and phenylarsine oxide (PhAsO) which mimic the effects of cyclophilin D for the PTP, while the channels were inhibited by BaCl_2 , MgCl_2 + ADP, and GdCl_2 , known inhibitors of the mitochondrial PTP. The authors also showed that the Ca^{2+} + Bz-423 induced channel activity was not due to the ADP/ATP translocase 1 or the VDAC1 & 2 contamination because the channel activity was not attenuated by the ADP/ATP translocase inhibitor bongkrekic acid or the VDAC inhibitor König's polyanion.

The most impressive experimental result was when the authors separated monomers, dimers, and tetamers and incorporated them into lipid films. On exposure to Ca^{2+} + Bz-423, the dimers and tetamers formed channels but the monomers did not (Fig. 2g).

Overall Analysis:

This paper provides convincing evidence that oligomeric complexes of the bovine F-ATP synthase can form Ca^{2+} -activated ion channels and that these channels have a number of physiological features in common with the mitochondrial PTP. The demonstration that the channel activity was not attenuated by bongkrekic acid or König's polyanion confirms that the F-ATP synthase preparation's channel function is not the product of ADP/ATP translocase or VDAC contamination, though an additional experiment with carboxyatractyloside could also have been included.

Answer. We agree that using also CATR would be useful, but given the number of experiments we had to perform in the revision and the limited amount of time for resubmission we have deferred these experiments.

Overall, I found the experiments convincing, specially that the F-ATP synthase can form Ca^{2+} -mediated ionic channels with characteristics associated with the PTP. Whether that pore is large enough to accommodate a 1500 Da molecule was not addressed. Regardless, I would support publication of this report without need for additional experiments.

Answer. Thank you very much for your support.

Minor Suggestions:

Legend Figure 2:

The sentence currently reads: "From the left: current traces before and after addition of 5 mM BaCl_2 (**c**, $[\text{Ca}^{2+}] = 3$ mM, $G_{\text{max}} = 500$ pS), 1 mM MgCl_2/ADP (**d**, $[\text{Ca}^{2+}] = 0.3$ mM, $G_{\text{max}} = 600$ pS), and 1 mM GdCl_3 (**e**, $[\text{Ca}^{2+}] = 0.66$ mM, $G_{\text{max}} = 1300$ pS)."

Recommended rephrasing: "From the left: current traces before and after addition of (**c**) 5 mM BaCl_2 ($[\text{Ca}^{2+}] = 3$ mM, $G_{\text{max}} = 500$ pS); (**d**) 1 mM MgCl_2/ADP ($[\text{Ca}^{2+}] = 0.3$ mM, $G_{\text{max}} = 600$ pS); and (**e**) 1 mM GdCl_3 ($[\text{Ca}^{2+}] = 0.66$ mM, $G_{\text{max}} = 1300$ pS)."

Answer. Done, thanks.

Question: Why were different levels of Ca^{2+} used in the three experiments different: $[\text{Ca}^{2+}] = 3$ mM; $[\text{Ca}^{2+}] = 0.3$ mM; $[\text{Ca}^{2+}] = 0.66$ mM?

Answer. Opening is not always observed at the same $[Ca^{2+}]$, and we chose to study the activity at the lowest Ca^{2+} concentration causing full opening.

Page 2:

Sentence currently reads: "The preparation displayed the correct subunit stoichiometry and revealed the presence of negligible contaminating proteins (Extended Data, Tables 2,3 and Fig. 2)." Questions: Is Extended Fig. 2 correct here? Why is the ratio of α and β subunits not 3X the axial subunits?

Answer. Reference to Figure 2 was wrong, the actual reference should have been to main Figure 1. About the ratio, the reference value of 100% for α subunit is an experimentally determined internal value, and may lead to overestimate the abundance of other subunits that are easier to detect by LC-MS/MS. We have removed the mention to "correct stoichiometry", thank you for pointing this out.

Page 4:

Sentence reads: "Moreover, mean and maximal conductance increased as a function of the Ca^{2+} concentration (Fig. 3b) suggesting that Ca^{2+} may contribute to channel assembly and/or trigger specific conformational changes."

Question: Should (Fig. 3b) actually be (Fig. 3 e & f)?

Answer. Thank you for pointing this out, this has now been corrected (with reference to the new Figure, which is now 5).

Extended Table 2:

Sentence reads: This has been detected only twice in its original form as a mono-charged specie (www.peptideatlas.org),

Recommendation: add an "s" to "specie" to give species

Answer. Done, thanks.

Reviewers' Comments:

Reviewer #1:

Remarks to the Author:

The authors have written responses and have altered some figures in the manuscript. Publication is still recommended, but there is an important issue regarding the claim that the channel activity is formed by ATP synthase dimers. In response to the following reviewer query, "According to the cryo-EM image shown in Fig. 1f, reconstituted proteoliposomes are loaded with many ATP synthase molecules. How can the authors explain that they get single channel recordings in the experiment where they fuse these liposomes with the planar bilayers ..." The authors have responded:

"Answer. This comment (also made by the second Reviewer) has been extremely useful... We have assessed the distribution of F-ATP synthase in the liposomes, and found that it associates with small liposomes..."

But this turns out to be related to another reviewer query:

"How can the authors be sure that the dimers remain intact when they fuse either liposomes or eluted protein with the planar membrane? Would it not be the case that the angle for dimer formation would be incorrect in a planar membrane?"

Because, as the authors now find, only very small liposomes have a probability of housing the ATP synthase, it seems probable that the small liposomes contain a tight enough angle to contain dimers, but that once these fuse with the planar membrane, the dimer must dissociate; the angle of the planar membrane, as opposed to the small liposome, would not support the continued dimeric structure. In addition, they reference their 2013 report in which running a gel of the reconstituted ATP synthase reveals equal bands for monomer and dimer, therefore at least 66% of the material reconstituted in the membrane is in monomeric form when they are recording "dimers".

Therefore, the authors' claim that the dimers only and not the monomers make the channel cannot be substantiated by their data, even more so now that they show that only small-angle liposomes harbor the dimeric structure. I recommend toning down the dimer language in the manuscript.

Reviewer #2:

Remarks to the Author:

Authors addressed my concerns and manuscript is now much improved.

The only issue that remains not entirely clear to this referee is authors interpretation of liposomes incorporation experiments. As indicated on figure 1i each liposome should contain several dozens of active ATPases. Thus even single liposome-bilayer fusion event (which is all-or-nothing) would incorporate a large number of active enzymes into the bilayer yet mostly single channel activity was observed. The most plausible interpretation of this data is that only small sub-population of ATPase can be converted into the pore. This possibility is particularly important to consider taking into account that in native patch-clamp PTP is a rare event while inner membrane is fairly tightly packed with ATPase proteins. Thus, if any (most of) ATPase population could be converted into PTP then why addition of Ca to the patch solution in native membranes doesn't cause multiple channels but only induces single channel activity. I believe it is important to discuss this issue since it might provide important guide in future search of the "pore" molecular structure within ATPase.

On a relevant subject. Control experiments with gramicidin D is worthy addition to the manuscript but they not necessarily prove the point. Proteo-liposomes method of protein channel reconstitution is a powerful technique for single channel recordings with the main trick to adjust protein concentration in such a way that 1 or less channels per liposome are present. This dilution is unique for each channel (and even different preparations of the same channel actually). Thus

results obtain for one channel can not be compared to the entirely different prep.

Reviewer #3:

Remarks to the Author:

The revised version of the manuscript addresses all the points raised by the reviewer and the reviewer is very happy with all these changes made. One particularly estimates also the added discussion about which subunits are (potentially) involved in the actual formation of the channel. This is a very contemporary point because of the recently published tetrameric, IF1-bound ATP synthase structure. The revised manuscript has gained strength and power to convince. Reading the discussion was a bit of a "bumpy road" and might still profit from some improvement of language, flow and the way the arguments were made, especially with respect to people not working in the field and not being aware of discussions from meetings. I would also recommend removing specific PIs (e.g. Walker). I liked the revised title of the work. The reviewer recommends its publication.

Reviewers' comments:

Reviewer #1 (Remarks to the Author):

The authors have written responses and have altered some figures in the manuscript. Publication is still recommended, but there is an important issue regarding the claim that the channel activity is formed by ATP synthase dimers. In response to the following reviewer query, "According to the cryo-EM image shown in Fig. 1f, reconstituted proteoliposomes are loaded with many ATP synthase molecules. How can the authors explain that they get single channel recordings in the experiment where they fuse these liposomes with the planar bilayers ..." The authors have responded:

"Answer. This comment (also made by the second Reviewer) has been extremely useful... We have assessed the distribution of F-ATP synthase in the liposomes, and found that it associates with small liposomes..."

But this turns out to be related to another reviewer query:

How can the authors be sure that the dimers remain intact when they fuse either liposomes or eluted protein with the planar membrane? Would it not be the case that the angle for dimer formation would be incorrect in a planar membrane?

Because, as the authors now find, only very small liposomes have a probability of housing the ATP synthase, it seems probable that the small liposomes contain a tight enough angle to contain dimers, but that once these fuse with the planar membrane, the dimer must dissociate; the angle of the planar membrane, as opposed to the small liposome, would not support the continued dimeric structure.

In addition, they reference their 2013 report in which running a gel of the reconstituted ATP synthase reveals equal bands for monomer and dimer, therefore at least 66% of the material reconstituted in the membrane is in monomeric form when they are recording "dimers".

Therefore, the authors' claim that the dimers only and not the monomers make the channel cannot be substantiated by their data, even more so now that they show that only small-angle liposomes harbor the dimeric structure. I recommend toning down the dimer language in the manuscript.

Answers.

- We agree that our study does not determine which oligomeric state the F-ATP synthase is in during PTP/MMC-like channel activity. We have re-written the relevant paragraph in the Discussion (lines 219-239) as follows:

"Native gel-eluted dimers and tetramers, but not monomers gave rise to currents after direct addition of the eluate to the *cis* side of the planar bilayer. The importance of this experiment lies in the fact that it further rules out the possibility that an unknown and difficult-to-detect contaminating protein may be responsible for the recorded channel activity. Indeed, a putative contaminating channel of a size similar to F-ATP synthase might have co-migrated with the dimer or with the oligomer, but not with both; and a contaminating channel with high affinity for F-ATP synthase should have co-migrated with the monomer as well. Hence, the possibility that a protein different from F-ATP synthase is the underlying cause of the

recorded currents becomes negligible. The absence of channel activity for the gel-eluted monomer and the equivalence of channel activity for the gel-eluted dimer and oligomer might suggest that the minimal channel forming unit is the F-ATP synthase dimer. There are, however, several caveats to this interpretation. For example, the gel elution process might have stripped off factors essential for channel formation (such as structural lipids) from the monomers but not from the dimers and oligomers; or dimers reconstituted into the bilayer might have self-assembled into oligomers before the actual channel formation. Indeed, self-oligomerization of purified F-ATP synthase after lipid bilayer reconstitution has already been demonstrated for the yeast dimer by combining the mild GRecon membrane reconstitution approach with cryo-electron tomography of the resulting proteoliposomes^{43,44}. Finally, channel formation itself might change the oligomeric state via monomerization. Thus, since we did not detect the oligomeric state and electric currents in the same experiment, the conformation of F-ATP synthase during PTP formation remains an open question.”

- The presence of monomeric F-ATP synthase after re-solubilization of proteoliposomes is now noted in the main text.
- Please note that we have mentioned the findings of the native gel-elution experiment in the abstract because this was specifically requested by reviewer #3.
- About the issue of how single-channel recordings can be reconciled with the presence of multiple F-ATP synthase complexes during current recording, please see our answer to reviewer #2.

Reviewer #2 (Remarks to the Author):

Authors addressed my concerns and manuscript is now much improved.

The only issue that remains not entirely clear to this referee is authors interpretation of liposomes incorporation experiments. As indicated on figure 1i each liposome should contain several dozens of active ATPases. Thus even single liposome-bilayer fusion event (which is all-or-nothing) would incorporate a large number of active enzymes into the bilayer yet mostly single channel activity was observed. The most plausible interpretation of this data is that only small sub-population of ATPase can be converted into the pore. This possibility is particularly important to consider taking into account that in native patch-clamp PTP is a rare event while inner membrane is fairly tightly packed with ATPase proteins. Thus, if any (most of) ATPase population could be converted into PTP then why addition of Ca to the patch solution in native membranes doesn't cause multiple channels but only induces single channel activity. I believe it is important to discuss this issue since it might provide important guide in future search of the “pore” molecular structure within ATPase.

Answer: We thank reviewer #2 for pointing out the importance of this issue, the proper discussion of which may indeed provide guidance to future studies on this question. We have added a new paragraph to the discussion (lines 240-251) which hopefully both conveys the importance of the issue and sheds some light on possible explanations:

“Oligomycin sensitivity and acidification of the internal volume indicate that close to all F-ATP synthase complexes incorporated in the liposomes exhibit their core function, i.e. coupled ATP hydrolysis and H⁺ pumping; and characterization of the proteoliposomes clearly shows insertion of many F-ATP synthase complexes per liposome. Therefore, the detection of single channel activity is surprising. A possible explanation is that only very few of the liposome-reconstituted F-ATP synthase complexes are MMC/PTP-competent, e.g. as a consequence of conformational heterogeneity. Alternatively, several complexes might participate in the measured currents with each single complex having only a low open probability, thus contributing to a few or even a single event during the total recording time. Irrespective of the underlying

mechanism, however, it should be noted that the single channel recordings reported here match the observation of single-channel MMC activity of patched mitoplasts⁴, which should contain a high-density of F-ATP synthase oligomers.”

On a relevant subject. Control experiments with gramicidin D is worthy addition to the manuscript but they not necessarily prove the point. Proteo-liposomes method of protein channel reconstitution is a powerful technique for single channel recordings with the main trick to adjust protein concentration in such a way that 1 or less channels per liposome are present. This dilution is unique for each channel (and even different preparations of the same channel actually). Thus results obtain for one channel can not be compared to the entirely different prep.

Answer. We agree, and indeed we had mentioned the gramicidin experiment as a tool to “explore the frequency of fusion events” (line 132 of the current manuscript). We have however also added “Although the two preparations obviously differ substantially” (lines 134-135) as a further cautionary statement.

Reviewer #3 (Remarks to the Author):

The revised version of the manuscript addresses all the points raised by the reviewer and the reviewer is very happy with all these changes made. One particularly estimates also the added discussion about which subunits are (potentially) involved in the actual formation of the channel. This is a very contemporary point because of the recently published tetrameric, IF1-bound ATP synthase structure. The revised manuscript has gained strength and power to convince.

Reading the discussion was a bit of a “bumpy road”; and might still profit from some improvement of language, flow and the way the arguments were made, especially with respect to people not working in the field and not being aware of discussions from meetings. I would also recommend removing specific PIs (e.g. Walker). I liked the revised title of the work.

The reviewer recommends its publication.

Answer. Thank you for the suggestion, the PI names have been removed and we have considerably simplified and shortened the relevant paragraph (current lines 267-280).